# The Accuracy Cost of Weakness: A Theoretical Analysis of Fixed-Segment Weak Labeling for Events in Time

**John Martinsson**                                                    *john.martinsson@ri.se*
*Computer Science*
*RISE Research Institutes of Sweden*
*Centre for Mathematical Sciences*
*Lund University*

**Tuomas Virtanen**                                                    *tuomas.virtanen@tuni.fi*
*Signal Processing Research Centre*
*Tampere University*

**Maria Sandsten**                                                    *maria.sandsten@matstat.lu.se*
*Centre for Mathematical Sciences*
*Lund University*

**Olof Mogren**                                                    *olof.mogren@ri.se*
*RISE Research Institutes of Sweden*
*Swedish Centre for Impacts of Climate Extremes (climes)*
*Climate AI Nordics*

**Reviewed on OpenReview:** *https://openreview.net/forum?id=tTw8wXBQ18*

## Abstract

Accurate labels are critical for deriving robust machine learning models. Labels are used to train supervised learning models and to evaluate most machine learning paradigms. In this paper, we model the accuracy and cost of a common weak labeling process where annotators assign presence or absence labels to fixed-length data segments for a given event class. The annotator labels a segment as "present" if it sufficiently covers an event from that class, e.g., a birdsong sound event in audio data. We analyze how the segment length affects the label accuracy and the required number of annotations, and compare this fixed-length labeling approach with an oracle method that uses the true event activations to construct the segments. Furthermore, we quantify the gap between these methods and verify that in most realistic scenarios the oracle method is better than the fixed-length labeling method in both accuracy and cost. Our findings provide a theoretical justification for adaptive weak labeling strategies that mimic the oracle process, and a foundation for optimizing weak labeling processes in sequence labeling tasks.

## 1 Introduction

In supervised machine learning, labeled datasets are required for training and evaluation. During evaluation, the accuracy of the labels determine the quality of the analysis. However, in practice, labels often contain noise that varies with the input sample and label type. Noisy training labels present a persistent challenge in machine learning (Liang et al., 2009; Song et al., 2022). Deep learning models, in particular, are prone to overfitting noisy labels, raising questions about the nature of generalization (Zhang et al., 2021). Regularization techniques such as dropout (Srivastava et al., 2014), data augmentation (Shorten & Khoshgoftaar,

2019), and weight decay (Krogh & Hertz, 1991) mitigate overfitting but fail to eliminate the performance gap between training on noisy versus clean labels (Song et al., 2022).

Beyond the well-documented challenges posed by noisy training labels, inaccurate evaluation labels present a significant, yet often overlooked, obstacle to reliable machine learning. When evaluation metrics are computed against noisy ground truth, the apparent "best" performing model might simply be the one that most closely reproduces the noise present in the evaluation set, rather than exhibiting superior generalization capabilities. This very issue, where noisy evaluation labels can lead to the rejection of models that have learned the true clean label distribution, is a central concern addressed by Görnitz et al. (2014). This can lead to the selection of suboptimal models that perform well on the flawed evaluation data but generalize poorly to unseen, cleaner data or data from real-world applications. Consequently, performance benchmarks can be inflated and misleading, hindering meaningful comparisons between different approaches. Therefore, understanding the characteristics of label noise, not just in the training data but also in the evaluation data, is crucial for developing and selecting models that are truly effective and robust.

Labels are typically obtained through human annotation, a process that involves significant time and financial investment, particularly for complex data like audio or time-series signals. In this work, we consider a form of weak labeling where the annotator assigns presence or absence labels to predefined data segments. This offers a practical and cost-effective approach for annotating large audio datasets (Martin-Morato & Mesaros, 2023). To reduce cost, weak labels avoid specifying precise boundaries within the data segments, focusing instead on general presence or absence of the target class. However, this simplification introduces noise into the labels, especially for data with time-varying characteristics, such as audio signals, where events can occur intermittently within the labeled segment (Turpault et al., 2021). Understanding and mitigating this noise is critical to effectively leverage weak labels in downstream applications (Kumar & Raj, 2016).

The noise in weak labels can be categorized into two types: class label noise (mislabeling event presence or absence in a segment) and segment label noise (mislabeling due to misaligned segment boundaries). While class label noise has been extensively studied (Song et al., 2022; Zhang et al., 2021), the effects of segment label noise remain underexplored. This type of noise significantly affects tasks such as sound event detection (Hershey et al., 2021; Turpault et al., 2021; Shah et al., 2018) and medical image segmentation (Yao et al., 2023). Strategies like pseudo-labeling (Dinkel et al., 2022), robust loss functions (Fonseca et al., 2019), and adaptive pooling operators (McFee et al., 2018) aim to address challenges when training on weak labels. However, fully understanding the impact of weak labels requires quantifying their accuracy (Shah et al., 2018; Turpault et al., 2021).

While adaptive annotation methods have been explored in some domains, fixed-segment (FIX) annotation remains the de facto standard for large-scale sound event datasets due to its simplicity and scalability. Foundational datasets such as AudioSet (Gemmeke et al., 2017a), CHIME (Foster et al., 2015b), OpenMIC-2018 (Humphrey et al., 2018a), YBSS-200 (Singh & Joshi, 2019), and SONYC (Bello et al., 2019), as well as more recent datasets such as VGG Sound (Chen et al., 2020), MATS (Morato & Mesaros, 2021), and MAESTRO Real (Morato et al., 2023) continue to rely on fixed segments. A practical motivation is that for many sound sources, precise boundaries are inherently ambiguous: a passing car or fading background noise lacks sharp onsets and offsets, and audio scenes frequently contain overlapping sources. These challenges make FIX annotation a practical default even when adaptive approaches are conceptually appealing.

In this paper we use the term weak labeling exclusively to refer to temporal ambiguity introduced by segment-level human annotation. We do not use the term in the sense of pseudo-labels or automatically generated annotations.

Current methods typically estimate label noise rates *after* collecting labels (Song et al., 2022), employing techniques like noise transition matrices (Li et al., 2021) or cross-validation (Chen et al., 2019). In contrast, predicting label noise rates *before* data collection remains largely unexplored. This is particularly challenging when the noise stems from human annotators, as it is difficult to formalize. In cases involving partially automated processes, however, the noise introduced by the automated component can often be modeled under specific assumptions.

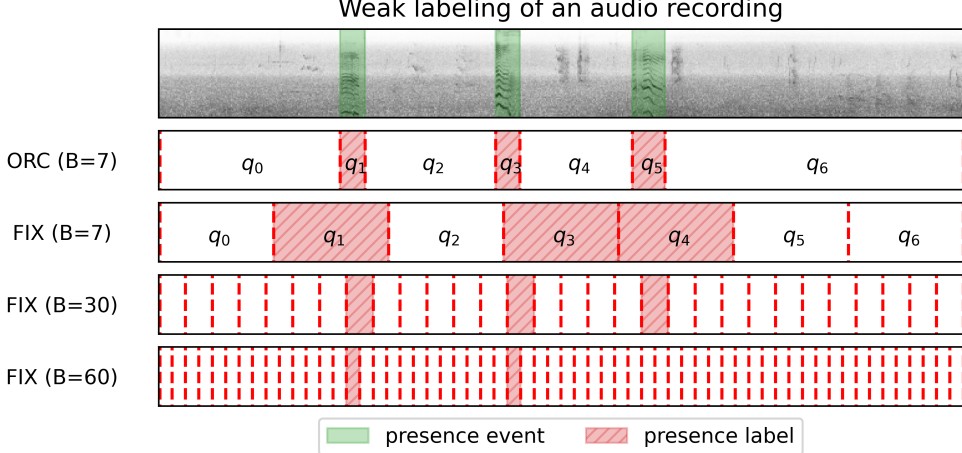

Figure 1: Resulting presence (red) and absence (white) labels from ORC and FIX weak labeling for an audio recording with three presence events (green). ORC weak labeling assigns labels to ground truth segments, achieving perfect label accuracy with $B = 7$ labels. FIX weak labeling, shown for different segment lengths ($B = 7$, $B = 30$, $B = 60$), introduces segment label noise as segments misalign with events. Longer segments reduce annotation cost but increase noise, while shorter segments align better but require more annotations. Note that too short segments ($B = 60$) may lead to the annotator missing the presence of the event because it does not cover a large enough fraction of it.

In this work, we model the automated component of a commonly used weak labeling method for segmentation tasks: fixed-length weak labeling (FIX). We quantify the segment label noise of this process, and study the expected label accuracy. This method, commonly employed in sound event detection, involves annotators providing presence or absence labels for fixed-length segments of the data (automated component), rather than specifying precise event boundaries. By simplifying the labeling process, FIX weak labeling reduces annotation effort but introduces segment label noise when segments misalign with the actual onsets and offsets of events. To benchmark this approach, we compare it to an oracle weak labeling method, ORC weak labeling, which assigns presence or absence labels to segments derived using the true onsets and offsets of the events.

Figure 1 illustrates the trade-offs between annotation cost and label accuracy for the ORC and FIX weak labeling methods. ORC weak labeling achieves perfect label accuracy by aligning the segments with the ground truth presence events (green) using a minimal number of annotated segments. In contrast, FIX weak labeling shows varying accuracy depending on segment length: shorter segments improve alignment ($B = 30$) but require more annotations, while longer segments ($B = 7$) reduce cost at the expense of accuracy. In addition, too short segments ($B = 60$) can lead to the annotator missing event presence. These trade-offs are central to understanding how FIX weak labeling can be used effectively. By analyzing the FIX weak labeling method, we provide a theoretical framework to guide data collection efforts.

Our analysis shows that FIX weak labeling systematically introduces segment noise, and we provide closed-form expressions for expected label accuracy and the optimal segment length. These results establish a quantitative baseline for annotation design, highlight the limits of non-adaptive strategies, and motivate the development of adaptive methods that approximate oracle labeling.

In summary, our contributions include:

- Closed-form expressions for label accuracy and annotation cost in FIX and ORC weak labeling, made tractable by assuming an annotator model and a simplified data distribution.

- A simulation study demonstrating that our theoretical framework generalizes to more complex data distributions and serves as an upper bound for the accuracy of FIX weak labeling.

- A theoretical foundation for developing adaptive weak labeling methods that better approximate ORC weak labeling, such as (Martinsson et al., 2024) for sound event detection and (Kim et al., 2023) for image segmentation.

Our analysis focuses on one-dimensional data, and the assumptions are justified by common characteristics in bioacoustic sound events. These time-localized, non-stationary animal vocalizations often require annotators to hear significant portions of the sound to assign accurate presence labels. Note, however, that while our framework is tailored to this domain, the principles extend to annotation of events in time in other data that shares these characteristics.

## 2 Problem Setting

The analysis is framed within a multi-pass binary labeling setting. Here, an annotator assigns binary labels (presence or absence) to data segments based on the occurrence of specific sound events. The annotator model abstracts how an annotator interacts with data by labeling segments, without requiring precise knowledge of event boundaries. While inspired by time-localized and non-stationary sound events, this framework is generalizable to any time series with similar characteristics.

It's important to emphasize that, in this weak labeling setting, the concept of overlapping events is not explicitly modeled. Overlapping events from the same class are treated as a single, longer presence event, because presence/absence labels cannot differentiate between individual event instances. For instance, in an audio recording with two birds calling simultaneously, this weak labeling framework simplifies the overlap into a single 'present' event. While this simplification is necessary when studying weak labeling in this setting, it fundamentally restricts our ability to resolve polyphony (the identification of multiple overlapping sound events). We leave the exploration of annotator models capable of providing richer labels to future work; this is beyond the scope of the current study.

For events of different classes occurring simultaneously, annotation is typically carried out in a multi-pass setup: each class is annotated independently, often by different annotators. Overlaps across classes are therefore not mutually exclusive—both events can be marked present within the same temporal window. The results from this paper are still valid when viewed for each class separately.

### 2.1 The Assumed Data Distribution

A sound event $e$ is defined by its start time $a_e \in \mathbb{R}$, end time $b_e \in \mathbb{R}$, and class $c_e \in \mathcal{C}$, denoted as $e = (a_e, b_e, c_e)$. Audio recordings are assumed to have finite length $T$, and we assume that the events are uniformly distributed locally in time (see Section 4.2 and Appendix A.4 for more details).

### 2.2 The Assumed Annotator Model

For a given sound event class $c \in \mathcal{C}$, the annotator decides the presence or absence of an event $e$ of class $c$ in a data segment $q = (a_q, b_q)$, where $d_q = b_q - a_q$ is the fixed-length of the segment. We will refer to $q$ as a query segment because it is queried for a presence or absence label. Let $l_q \in \{0, 1\}$ denote the weak label indicated by the annotator for query segment $q$, where $l_q = 1$ indicates presence of an event of class $c$ in $q$ and $l_q = 0$ indicates absence of that event class in $q$. Detecting the presence of an event requires observing a sufficient fraction of the event within the query segment, formalized as follows:

**Definition 1.** The *event fraction* is the fraction of the total event duration $d_e = b_e - a_e$ that overlaps with the query segment $q$,

$$h(e, q) = \frac{|e \cap q|}{d_e}, \tag{1}$$

where $e \cap q$ is the intersection of $(a_e, b_e)$ and $(a_q, b_q)$.

**Definition 2.** The *presence criterion* $\gamma \in (0, 1]$ is the minimum event fraction required for the annotator to detect the presence of $e$ in $q$,

$$h(e, q) \geq \gamma. \tag{2}$$

The annotator assigns a presence label ($l_q = 1$) to $q$ if there is sufficient overlap with any presence event $e$ of class $c$ ($h(e, q) \geq \gamma$); otherwise, it assigns an absence label ($l_q = 0$).

The parameter $\gamma$ reflects the annotator's sensitivity: lower $\gamma$ values indicate sensitivity to smaller event fractions, while higher values require larger fractions. This model of perceptual ability is particularly suited for non-stationary events (e.g., a specific birdsong) where recognizing a relative portion of the event's structure is key, as opposed to stationary sounds (e.g., an engine hum) which might be identified after a fixed absolute duration.

This framework captures variability in annotator behavior. For example, detecting "human speech" or "bird song" may only require hearing a small fraction of the event ($\gamma$ closer to 0), while recognizing specific phrases or bird species might demand a near-complete observation ($\gamma$ closer to 1). The value of $\gamma$ thus depends on the annotator and the complexity of the event class. This model provides a flexible yet precise way to simulate annotator behavior and quantify their labeling performance. However, it is important to note that this model is deterministic, focusing on temporal alignment between events and the query segment. In practice, human annotation often involves stochastic factors, such as variability in perception and judgment, which are not explicitly modeled here.

### 2.3 Label Accuracy

Label accuracy measures the alignment between annotator-provided labels and ground truth labels:

**Definition 3.** The label accuracy is defined as

$$F(e, q, \gamma) = \begin{cases} \frac{|e \cap q|}{d_q}, & \text{if } l_q = 1, \\ \frac{d_q - |e \cap q|}{d_q}, & \text{if } l_q = 0. \end{cases} \tag{3}$$

For instance, consider a 3-second query segment ($d_q = 3$) that overlaps exactly one second ($|e \cap q| = 1$) with a 2-second sound event ($d_e = 2$) of the class bird song ($c =$ "bird song"). The annotator assigns a presence label ($l_q = 1$) with label accuracy $\frac{|e \cap q|}{d_q} = \frac{1}{3}$ if half or less of the event needs to be in the query segment ($\gamma \leq 0.5$). Contrary, the annotator assigns an absence label ($l_q = 0$) with label accuracy $\frac{d_q - |e \cap q|}{d_q} = \frac{3-1}{3} = \frac{2}{3}$ if more than half of the event ($\gamma > 0.5$) needs to be in the query segment. This formulation isolates the segment label noise ($1 - F(e, q, \gamma)$) introduced by the automated component (fixed-length segments) of the FIX weak labeling method.

## 3 The Label Accuracy and Cost of ORC Weak Labeling

Let us start with the ORC weak labeling method. This method uses a priori information about the event start and end times and is therefore not available in practice, but should be seen as an upper bound on what can be achieved with weak labeling. The start and end times of the true presence and absence events are used to construct the query segments:

$$\mathbb{Q}_{\text{ORC}} = \{(a_0, b_0), (a_1, b_1), \ldots, (a_{B_{\text{ORC}}-1}, b_{B_{\text{ORC}}-1})\} = \{q_0, \ldots, q_{B_{\text{ORC}}-1}\}, \tag{4}$$

where $(a_i, b_i)$ is the $i$th ground truth presence or absence event. The annotator indicates presence or absence for each of these segments, which by construction results in the ground truth annotations, illustrated in Figure 2. In the example, there are three target events (green), and four absence events, which means that $B_{\text{ORC}} = 7$. In general $B_{\text{ORC}} \in \{2M - 1, 2M + 1\}$, where $M$ denotes the number of presence events. The number of absence events can be fewer than $2M + 1$ if the recording starts or ends with a presence event, however, for simplicity and without losing generality, we will consider $B_{\text{ORC}} = 2M + 1$ as the minimum number of query segments needed for ORC to derive the ground truth. From an annotation cost perspective, this is the most cautious choice, and it is also the most likely outcome. The query accuracy is 1 for each query segment since by construction the fraction of correctly labeled data in each query segment will be 1 when given the correct presence or absence labels.

Figure 2: ORC weak labeling of an audio recording with three target events ($M = 3$) shown in green and four absence events. The $B_{\text{ORC}} = 7$, query segments $q_0, \ldots, q_6$ are derived from the ground truth segmentation of the data, and therefore the label accuracy will by definition be 1.

In summary, the ORC weak labeling method produces annotations with label accuracy 1, using the minimum number of query segments needed to achieve this. We use this as a reference on what can be achieved for weak labeling data.

## 4    The Label Accuracy and Cost of FIX Weak Labeling

The outline of this section is as follows. In Section 4.1 we define the FIX labeling method. In Section 4.2 we derive a closed-form expression for the expected label accuracy of a query segment given that it overlaps with a single event of deterministic event length. We note that it is only in the cases of overlap between a query segment and an event that a presence label can occur under the assumed annotator model, and that the expectation in label accuracy over these cases therefore can be viewed as the expected presence label accuracy. For the remainder of the paper we will simply write expected label accuracy when referring to the expectation over the overlapping cases, unless explicitly stated otherwise.

In the same section we derive the optimal query length with respect to the expected label accuracy, the maximum expected label accuracy and the number of query segments needed (proxy for annotation cost). In Section 4.3 we explain how the expression for expected label accuracy can be used in the case of a single event of stochastic length, and in Section 4.4 we explain under which conditions this can be used when multiple events can occur. Finally, we derive a closed form expression for the expected label accuracy of an audio recording with multiple events of stochastic length in Section 4.5, and provide an alternative interpretation of the theory in Section 4.6.

### 4.1    The FIX Weak Labeling Method

The FIX weak labeling method, commonly used in practice, splits the audio recording into fixed and equal length query segments, and then an annotator is asked to provide either a presence or absence label for each of the query segments. Let $B_{\text{FIX}}$ denote the number of query segments used, then the query segments for an audio recording of length $T$ are defined as

$$\mathbb{Q}_{\text{FIX}} = \{(a_0, b_0), (a_1, b_1), \ldots, (a_{B_{\text{FIX}}-1}, b_{B_{\text{FIX}}-1})\} = \{q_0, \ldots, q_{B_{\text{FIX}}-1}\}, \tag{5}$$

where the start and end timings of each query segment is $q_i = (a_i, b_i) = (id_q, (i+1)d_q)$ and the fixed query segment length is $d_q = T/B_{\text{FIX}}$. We illustrate this in Figure 3, where the presence criterion for the annotator is $\gamma = 0.5$. There are three presence events and four absence events, and using only $B_{\text{FIX}} = 7$ query segments results in annotations with an average label accuracy that is lower than 1.

We want to find an expression for the expected label accuracy for a given data distribution and query segment length. In addition, we want to understand the query length that maximize the expected label accuracy.

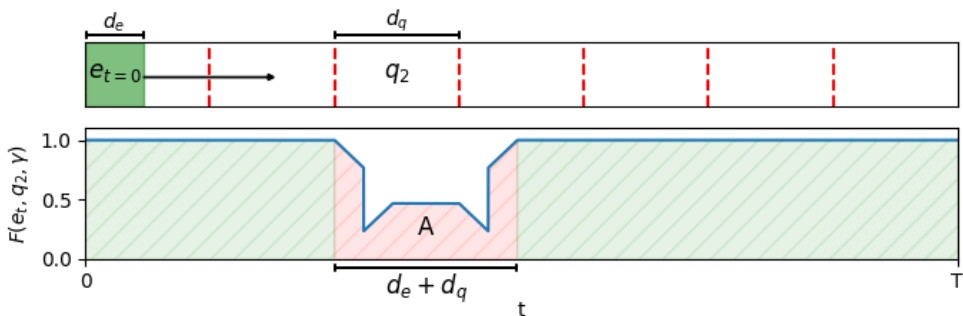

Figure 3: Illustration of the FIX weak labeling method. The audio recording contains presence events (green). The FIX method divides the recording into fixed-length query segments (e.g., $q_0$ to $q_6$). Note how the alignment between segments and presence events affects the accuracy of presence labels (red hatched).

Figure 4: *Top panel:* A single event ($e_t$) of length $d_e$ can occur at various end times ($t$) within the recording of length $T$. *Bottom panel:* The resulting label accuracy for query segment $q_2$ (arbitrarily chosen for illustration) of length $d_q$ as a function of the event's end time ($t$). Overlap between the event and the query segment leads to segment label noise and a reduced label accuracy, which in this case occur when $t \in [a_2, a_2 + d_e + d_q]$ where $a_2$ is the start time of $q_2$. The red hatched area ($A$) represents the cumulative label accuracy during these overlapping scenarios. The figure illustrates the evolution of the accuracy function $F(e_t, q, )$ as the event end time sweeps across the segment. The apparent central alignment at 0.5 is specific to this example and should not be interpreted as a general property of FIX labeling.

## 4.2 The Expected Label Accuracy of a Query Segment given Event Overlap

To derive a tractable closed-form expression, we analyze a simplified data distribution where each recording of length $T$ contains a single event of deterministic length $d_e$. This idealized case is the simplest possible annotation scenario, and the resulting accuracy can therefore be interpreted as a theoretical upper bound: any added complexity, such as multiple events or variable event lengths, introduces additional opportunities for error.

The setup is illustrated in the upper panel of Figure 4, where a single event $e_t$ of length $d_e$ can occur at any time $t \in [0, T]$ (indicated by the arrow). The bottom panel of Figure 4 shows the label accuracy for a specific query segment ($q_2$) as the end time ($t$) of the event varies. The area ($A$) highlighted in hatched red indicates the label accuracy in the cases of overlap between the query segment and the event, and the area in hatched green indicate the label accuracy in the cases of no overlap, which is by the definition of the annotator model is always 1. Crucially, while this figure illustrates the accuracy for query segment $q_2$, the shape of this accuracy function remains the same for other query segments; only its position along the x-axis would change.

To simplify the mathematical analysis, without loss of generality, we can fix the query segment to start at time 0, $q = (0, d_q)$, and represent the event with its ending time $t$ as $e_t = (t - d_e, t)$. In this way, $t \in [0, d_e + d_q]$ describes all possible overlap occurrences. That is, when $t = 0$ the event ends at the start of the query segment, and when $t = d_e + d_q$ the event starts at the end of the query segment. To formalize this, we can express the expected label accuracy in case of overlap by integrating over all possible event end times ($t$) where overlap occurs:

$$\mathbb{E}_{t \sim p}\left[F(e_t, q, \gamma)\right] = \int_0^{d_e + d_q} F(e_t, q, \gamma) p(t) \mathrm{d}t \tag{6}$$

$$= \frac{1}{d_e + d_q} \int_0^{d_e + d_q} F(e_t, q, \gamma) \mathrm{d}t \tag{7}$$

$$= \frac{A}{d_e + d_q}, \tag{8}$$

where $t \sim p$ denotes a random variable t distributed according to a distribution $p$, and $p(t)$ denotes the probability of realization $t$. We assume that the distribution of the relative offsets $t$ between events and overlapping query segments is uniformly distributed (empirically verified in Appendix A.4). There are two sources of variation that makes this plausible: (i) the start time of the recording varies depending on when the recording session was started, and (ii) the start time of the event varies depending on when the sound source emits the event. Note that this assumption is likely to not hold if $d_q \gg d_e$, but that leads to very weak labels which is not wanted in practice. Using this assumption we get $p(t) = 1/(d_e + d_q)$, and by observing that the integral $\int_0^{d_e + d_q} F(e_t, q, \gamma) \mathrm{d}t$ describes the hatched red area denoted $A$ in Figure 4 we arrive at the final expression in Eq. 8.

Remember that absence labels can occur when there is no overlap (always correct) and when there is overlap but the presence criterion is not fulfilled, and presence labels can only occur when there is overlap and the presence criterion is fulfilled. Therefore, inaccurate labels only occur in the case of overlap. The expected label accuracy in the case of overlap therefore describes the accuracy of the labels when segment label noise can occur, which happens around the boundaries of the true event.

In Appendix A.1 we show how to express $A$ in terms of the event length $d_e$, the query segment length $d_q$ and the presence criterion $\gamma$ under the assumption that the annotator presence criterion can be fulfilled ($d_q \geq \gamma d_e$), and that it can not be fulfilled ($d_q < \gamma d_e$). Finally, we arrive at the following four main theorems:

**Theorem 1.** The expected label accuracy in case of overlap between a query segment $q$ of length $d_q$ and a single event $e$ of deterministic length $d_e$ is

$$f(d_q) = \mathbb{E}_{t \sim p}\left[F(e_t, q, \gamma)\right] = \begin{cases} \frac{d_e \left(2\gamma d_q - 2\gamma^2 d_e + d_q\right)}{d_q (d_e + d_q)}, & \text{if } d_q \geq \gamma d_e, \\ \frac{d_q}{d_e + d_q}, & \text{if } d_q < \gamma d_e, \end{cases} \tag{9}$$

when the presence criterion for the annotator is $\gamma$.

*Proof.* See Appendix A.1 for the proof. We show how to express the area $A$ in Eq. 8 in terms of $d_e$, $d_q$ and $\gamma$ for the two assumptions: $d_q \geq \gamma d_e$, and $d_q < \gamma d_e$. □

**Theorem 2.** The query length that maximizes the expected label accuracy in case of overlap for a given event length $d_e$ is

$$d_q^* = d_e \gamma \frac{2\gamma + \sqrt{4\gamma^2 + 4\gamma + 2}}{2\gamma + 1}. \tag{10}$$

*Proof.* See Appendix A.2 for the proof. We compute the derivative of $f(d_q)$ with respect to $d_q$, and show that $d_q^*$ is the maximum. □

**Theorem 3.** The maximum expected label accuracy in case of overlap between a query segment of length $d_q$ and an event of length $d_e$ when $d_q \geq \gamma d_e$ is

$$f^*(\gamma) = f(d_q^*) = 2\gamma\left(2\gamma + 1 - \sqrt{4\gamma^2 + 4\gamma + 2}\right) + 1. \tag{11}$$

*Proof.* See Appendix A.3 for the proof. We substitute $d_q$ for $d_q^*$ in Eq. 9. □

**Theorem 4.** The number of queries $B_{\text{FIX}}^*$ (cost) that are needed by FIX to maximize the expected label accuracy in case of overlap for an audio recording of length $T$ when $d_e = 1$ is

$$B_{\text{FIX}}^* = \frac{T}{d_q^*}. \tag{12}$$

*Proof.* $T/B_{\text{FIX}}^* = d_q^*$, which by Theorem 2 leads to maximum label accuracy. □

In summary, Theorem 1 gives us an expression $f(d_q)$ for the expected label accuracy when query segments of length $d_q$ are used to detect events of length $d_e$ and the presence criterion for the annotator is $\gamma$. We use this to find the query segment length $d_q^*$ that maximize the expected label accuracy, leading to Theorem 2. Theorem 2 show the query segment length $d_q^*$ that maximizes expected label accuracy for a given event length and annotator criterion. Further, by inserting $d_q^*$ into Theorem 1, $f^*(\gamma) = f(d_q^*)$, we get Theorem 3, which is the maximum achievable expected label accuracy for a given annotator criterion $\gamma$. We have omitted the case $d_q < \gamma d_e$ when deriving $f^*(\gamma)$, since maximizing the expected label accuracy in the case when the annotator presence criterion can not be fulfilled is not very interesting, since we can not get presence labels. Note that $f^*(\gamma)$ is a function of only $\gamma$, meaning that the maximum expected label accuracy is independent of the target event length when considering a single deterministic event. Finally, Theorem 4 show that an annotator needs to weakly label $B_{\text{FIX}}^*$ query segments for each audio recording to achieve the maximum label accuracy in expectation, which can be seen as a proxy for annotation cost.

There is arguably no simpler audio data distribution to annotate than when recordings only contain a single event of deterministic length (except for when no event occurs at all). We can therefore treat $f^*(\gamma)$ as an upper bound on the maximum expected label accuracy for any audio distribution. We demonstrate this empirically in the results in Section 6. However, in practice audio recordings often contain events that vary both in length and number. Let us therefore consider how the derived theory can be useful also in these cases.

### 4.3 Stochastic Event Length

Events may vary in length according to some event length distribution. Let $p(d_e)$ denote the probability of the outcome that an event has length $d_e$, and let $d_e \sim p(d_e)$ denote that $d_e$ is a sample from that distribution. The expected label accuracy over a distribution of event lengths for a given $\gamma$ and query segment length $d_q$ can then be computed as

$$\mathbb{E}_{d_e \sim p(d_e)}[f(d_q)] = \int_0^\infty f(d_q)p(d_e)\mathrm{d}d_e. \tag{13}$$

While we do not provide a closed form solution for this, we can solve the integral in Eq. 13 by numerical integration. Note that $d_q^*$ in Theorem 2 depends on the single event length $d_e$, and to find it for a distribution we would need to solve Eq. 13 for a range of $d_q$ and find the one that leads to the best label accuracy. However, for some event length distributions, setting $d_e$ to the average of the distribution turns out to be a good heuristic. We perform a simulation study in Section 6.1.2 to support these claims.

### 4.4 Multiple Events

There may be multiple ($M$) events present in a given audio recording. In Figure 5 we show the label accuracy for all possible occurrences of a query segment $q_t$ in a recording with two events ($M = 2$). Note that we have put the subscript $t$ on the query segment ($q_t$) instead of the event as in the prior analysis. This formulation is entirely equivalent, but when talking about multiple events it is more intuitive to consider them as fixed in time for a given recording, and that the query segments occur relative them at random. There are now two regions where overlap occurs, one around $e_1$ and one around $e_2$. On average we get $2A/2(d_e + d_q) = A/(d_e + d_q) = f(d_q)$. That is, the theory we derived for the single event case explains the multiple event case.

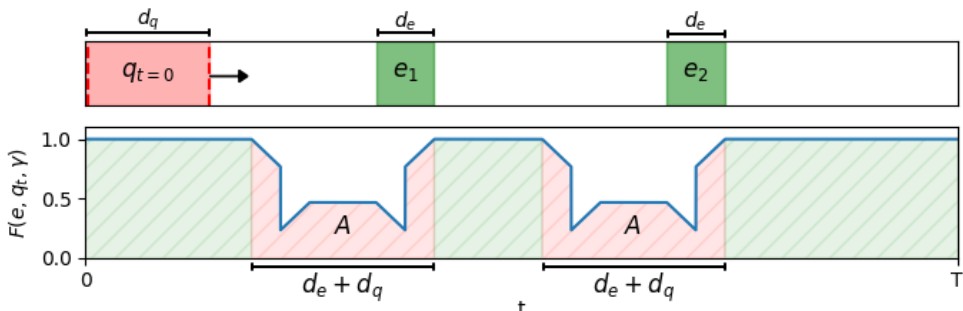

Figure 5: *Top panel:* Two events ($M = 2$) of length $d_e$ that are fixed in time within a recording of length $T$, and a query segment $q_t = (-d_q + t, t)$. *Bottom panel:* The resulting label accuracy of $q_t$ for $t \in [0, T - d_q]$, simulating that the $q_t$ can appear anywhere at random in time in relation to the events. As before, when there is overlap between the query segment and an event the label accuracy is below 1, otherwise it is always 1.

However, for this to hold we need to assume that for any event the closest other event is least $d_q$ away in time. In Figure 5 this holds since the start of $e_2$ is at least $d_q$ away from the end of $e_1$. If this assumption holds then the expected label accuracy for multiple events is $f(d_q)$. The assumption is plausible if events are sparse in relation to $d_q$. Note that $d_q^* \in (0, d_e \frac{2+\sqrt{10}}{3}]$ for $\gamma \in (0, 1]$ according to Theorem 2. That is, when considering the optimal query length $d_q^*$ this assumption translates to that events should be no closer than approximately $1.72 d_e$ for $\gamma = 1$, $0.81 d_e$ for $\gamma = 0.5$, and 0 for $\gamma \to 0$. We perform a simulation study in Section 6.1.3 to see the effect of breaking this assumption, and we leave it to future work to derive the expected label accuracy in case of overlap for multiple events.

## 4.5 The Expected Label Accuracy of an Audio Recording

We now know the expected label accuracy of a query segment given event overlap, and how to use this for a stochastic event lengths and multiple events. We can use this to derive an expression for the expected label accuracy of and audio recording of finite length ($T$) that has multiple ($M$) stochastic event lengths ($d_e \sim p(d_e)$).

**Theorem 5.** The expected label accuracy for an audio recording of length $T$, with $M$ events of stochastic event length $d_e \sim p(d_e)$ that are spaced at least $d_q$ apart is

$$\mathbb{E}_{d_e \sim p(d_e)} \left[ -\frac{2M d_e^2 \gamma^2}{T d_q} + \frac{2M d_e \gamma}{T} - \frac{M d_q}{T} + 1 \right]. \tag{14}$$

*Proof.* We will do this proof by picture. In Figure 5 we have two events ($M = 2$), in general for $M$ events the accumulated label accuracy in the cases of overlap is $MA$ (the sum of the hatched red areas), the total amount of overlapping cases is $M(d_e + d_q)$ and the total amount of non-overlapping cases is therefore $T - M(d_e + d_q)$ for an audio recording of length $T$. In the case of no overlap, the label accuracy is always 1, which means that the accumulated label accuracy in the case of no overlap (sum of the green hatched areas) is $T - M(d_e + d_q)$. Normalizing for the entire duration of the recording we arrive at

$$\frac{AM + T - M(d_e + d_q)}{T} = -\frac{2M d_e^2 \gamma^2}{T d_q} + \frac{2M d_e \gamma}{T} - \frac{M d_q}{T} + 1, \tag{15}$$

and as before we can simply compute an expectation over the event length distribution. $\square$

Theorem 5 tells us the expected label accuracy under FIX weak labeling with query segment length $d_q$ for an audio recording of length $T$, with $M$ events of stochastic event length $d_e \sim p(d_e)$. If we want to account

for class label noise, where the annotator gives the wrong label with probability $\rho$, this can be included by simply scaling the whole expression in Eq. 14 by $(1 - \rho)$. That is, the expected label accuracy for the cases of overlap allows us to express a variety of things about the expected label accuracy of an audio recording.

However, note that we have $T$ in the denominator of all terms except the term that is 1, meaning that if we let $T$ approach $\infty$, then the expected label accuracy approaches 1. That is, considering the accuracy of both absence and presence labels equally can lead to hiding the effect that we want to understand in this paper, which is the effect of $d_q$ on the accuracy of the presence labels. We could derive a balanced accuracy in a similar way as above, but instead we choose to continue our analysis looking only at the expected label accuracy in the case of overlap.

### 4.6 Expected Label Accuracy given Overlap when $d_q = \delta d_e$

As a result of the proof for Theorem 3 in Appendix A.3 we get an alternative dimensionless interpretation of the expected label accuracy when the query segment length is expressed as a factor of the event $d_q = \delta d_e$,

$$f(\delta d_e) = \frac{(2\gamma + 1)\delta - 2\gamma^2}{\gamma(1 + \gamma)}, \tag{16}$$

and an expression for the ratio that maximizes it

$$\delta^* = \frac{d_q^*}{d_e} = \gamma \frac{2\gamma + \sqrt{2\gamma^2 + 2\gamma + 1}}{2\gamma + 1}. \tag{17}$$

This alternative formulation illustrates that it is the ratio $\delta = d_q/d_e$ that affects the expected label accuracy of a single event, and not the absolute lengths $d_q$ and $d_e$. Further, we can use this interpretation to rewrite Theorem 5 as

$$\mathbb{E}_{\delta \sim p(\delta)}\left[\frac{Md\delta\left(-\delta + 2\gamma\right) - 2Md\gamma^2 + T\delta}{T\delta}\right], \tag{18}$$

where $\delta$ denotes a random variable with probability distribution $p(\delta)$.

## 5 Simulating the Label Accuracy of FIX Weak Labeling

To validate the theory, we simulated FIX labeling of various audio recording distributions and compared the average simulated label quality with the theoretical results from Section 4.2. The code used for these simulations is released openly[1].

We generated 1000 audio recordings of length $T = 100$ seconds for each configuration. The number of events, $M$, and the event length distributions varied across simulations, as detailed below:

- **Single Event with Deterministic Length:** We simulated recordings with $M = 1$ event of deterministic length $d_e = 1$ second.

- **Single Event with Stochastic Length from Normal Distributions:** We drew event lengths from two normal distributions with the same mean but different variances ($\mathcal{N}(3, 0.1)$ and $\mathcal{N}(3, 1)$), and from two normal distributions with different means but the same variance ($\mathcal{N}(0.5, 0.1)$ and $\mathcal{N}(5, 0.1)$). For these simulations, $M = 1$.

- **Single Event with Stochastic Length from Gamma Distributions:** We sample event lengths from two gamma distributions (offset by 0.5 seconds due to computation cost) with different shape parameters but the same scale parameter ($\mathrm{Gamma}(0.8, 1) + 0.5$ and $\mathrm{Gamma}(0.2, 1) + 0.5$) with $M = 1$.

- **Single Event with Stochastic Length from Real Length Sample:** We used the event length distributions for dog barks and baby cries from the NIGENS dataset (Trowitzsch et al., 2019) with $M = 1$.

---

[1]https://github.com/johnmartinsson/the-accuracy-cost-of-weakness

- **Multiple Events with Deterministic Length:** We simulated recordings with multiple events ($M = 30$ and $M = 50$) where each event had a deterministic length of $d_e = 1$ second.

For recordings with stochastic event lengths or multiple events, the length of each of the $M$ events was sampled from the specified distribution. Each sampled event was then placed randomly within the recording. The start time $a_e$ of each event was drawn uniformly at random from $[0, T - d_e]$. If multiple events were present, overlapping events were merged into one presence event. For each generated audio recording, we simulated FIX labeling using different annotator presence criteria $\gamma \in [0.01, 0.99]$ and a range of query segment lengths $d_q$. The query segment lengths were linearly spaced between a small fraction of the minimum event length observed in the distribution and a value several times the maximum observed event length.

We then computed the average label accuracy over the query segments that overlaps with an event in each recording. For each query segment $q$ we check if the annotator presence criterion ($h(e, q) \geq \gamma$) is fulfilled for any event $e \in E$, where $E$ is the set of all events that overlap with $q$. If this is true for any of the events then $q$ is given a presence label ($l_q = 1$) otherwise it is given an absence label ($l_q = 0$). The label accuracy is then computed in a similar way as in Eq. 3, but since we can now have multiple events overlapping with the same query segment, we need to consider the union of all overlapping events $\cup_{e \in E} e$ when computing the label accuracy of assigning label $l_q$ to that query segment. The total amount of overlap becomes $|(\cup_{e \in E} e) \cap q|$ instead of $|e \cap q|$. However, when $M = 1$ this is equivalent to Eq. 3 ($|(\cup_{e \in E} e) \cap q| = |e \cap q|$), since $|E| = 1$.

In this way, we simulated the effect of breaking the assumption that events are spaced at least $d_q$ apart, and could better understand the effect this had when compared to the derived theory. Finally, for each considered $\gamma$, we empirically determined the maximum average label accuracy across all tested query lengths and the corresponding optimal query length. These empirical results were then compared to the theoretical predictions.

# 6 Results

In this section we present the results of the simulated annotation process, and show how these connect to the derived theory. We start by looking at the expected label accuracy and the query segment length that maximize the expected label accuracy for FIX and ORC weak labeling, and then we relate this to the annotation cost.

## 6.1 Expected Label Accuracy given Overlap

We evaluate how different annotator presence criteria ($\gamma$) influence the achievable label accuracy given overlap under FIX weak labeling. We first examine the case of a single event with a deterministic length, then extend our simulation study to stochastic event lengths, and finally to multiple events occurring within the same recording.

### 6.1.1 Single Event with Deterministic Length

The simulated results are derived using the simulation setup described in section 5, with $M = 1$ (a single event) and $d_e = 1$ (deterministic length). In Figure 6, we show the maximum expected label accuracy given overlap (left) and the corresponding query length that maximize the label accuracy (right) for different $\gamma$. $f^*(\gamma)$ is the maximum expected label accuracy achievable with annotator presence criterion $\gamma$ for the considered event length. We can see that the simulated average label accuracy closely follows the expected label accuracy, and that the corresponding segment length leading to this maximum is the same in theory and simulation.

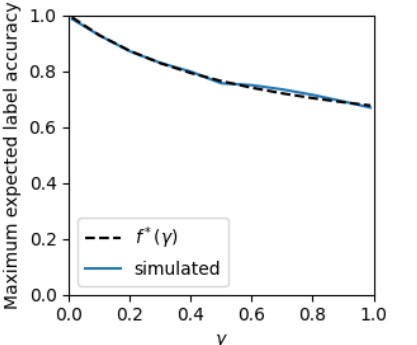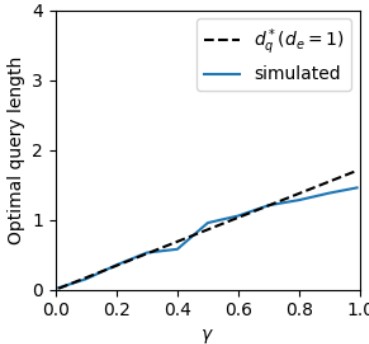

Figure 6: In the left panel we show the maximum expected label accuracy, $f^*(\gamma)$, for different $\gamma$, and the average maximum label accuracy from the simulations. In the right panel we show the query length that leads to this maximum label accuracy in theory, for $d_e = 1$, and in simulation. The theory follows the simulations well.

In Figure 6 we see that if the annotator needs to hear more than 50% of the sound event to detect presence ($\gamma = 0.5$) then the highest achievable label accuracy is $f^*(0.5) \approx 0.76$. This means that on average there is around 34% segment label noise around the presence labels. We also see that the query length that gives the maximum label accuracy is $d_q^* \approx 0.81$. The gap to the ORC weak labeling method which always gives a label accuracy of 1, is large especially for large $\gamma$. In general, we can see how the maximum label accuracy deteriorates with a growing $\gamma$, and which query segment length to choose to maximize label accuracy in expectation.

### 6.1.2 Single Event with Stochastic Length

We now consider stochastic event lengths. We do this to better understand the effect of the event length distribution on the maximum expected label accuracy and the optimal query length. We solve the integral in Eq. 13 by numerical integration over different event length distributions, and compare with the theory derived for a single deterministic event length and simulations. In each figure we present the derived theoretical rules $f^*(\gamma)$ and $d_q^*$ for the simplified event length distribution, the results from integration of Eq. 13 with different event length distributions $p(d_e)$ (numerical), and the simulated results using the procedure described in section 5 (simulated) where event lengths are sampled from different distributions. Note that, since $d_q^*$ is derived for a deterministic event length $d_e$, and require a choice of this value, we set $d_e$ to the average event length ($\mu$) for each distribution in these experiments as a heuristic. We then present the maximum expected label accuracy for different $\gamma$ (left in figures) and the query segment length that maximizes the expected label accuracy (middle in figures), and the histogram for the considered event length distributions (right in figures).

In Figure 7 and Figure 8 we see that the mean and variance of the normal distribution have a small (if any) effect on the maximum expected label accuracy, but the mean does affect which query segment length that maximizes the expected label accuracy. We also see that $d_q^*$ follows the simulated and numerical optimal query length well for all considered normal distributions, when $d_e$ is set to the average event length ($\mu$) for the considered event length distribution. The average event length can be used as a heuristic value if we only know the average and not the true distribution to integrate over.

In Figure 9 we can see that a gamma distribution does affect the maximum expected label accuracy, and that simply setting $d_e$ to the average event length of the distribution leads to underestimating the optimal query length. Since it is not possible to optimize for both short and long events at the same time using FIX weak labeling, this type of distribution is quite challenging.

In Figure 10 we validate the theory against a real sample of event lengths from either baby cries or dog barks. Numerical integration between the derived expression and the histogram predicts the simulations well.

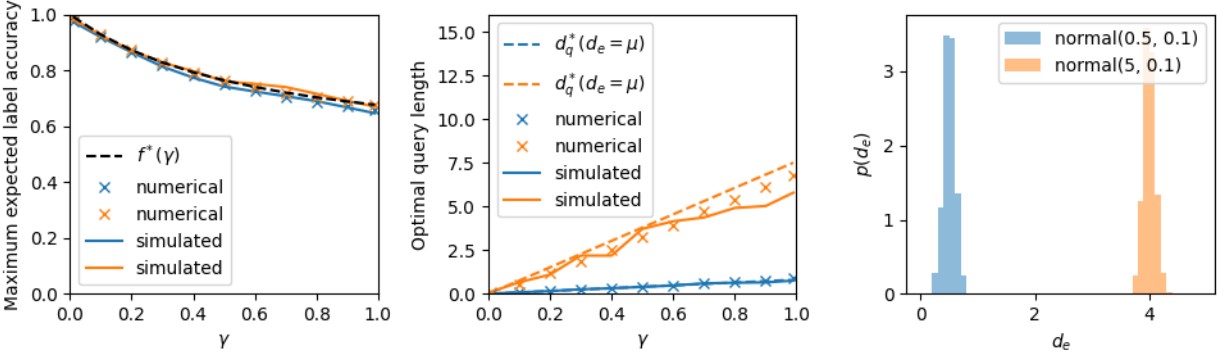

Figure 7: We validate the theory for stochastic event lengths drawn from two normal distributions with different means, but the same variance. We show the expected label accuracy (left panel), the optimal query length (middle panel), and the considered event length distributions (right panel).

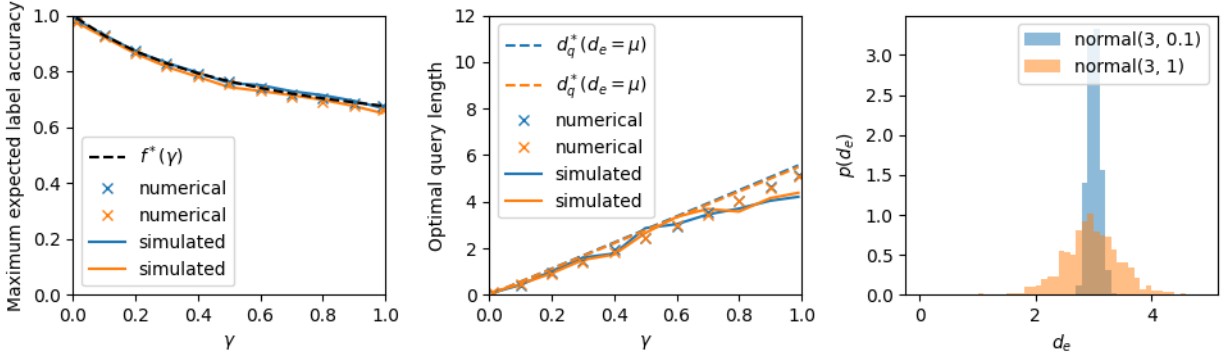

Figure 8: We validate the theory for stochastic event lengths drawn from two normal distributions with different variance, but the same mean. We show the expected label accuracy (left panel), the optimal query length (middle panel), and the considered event length distributions (right panel).

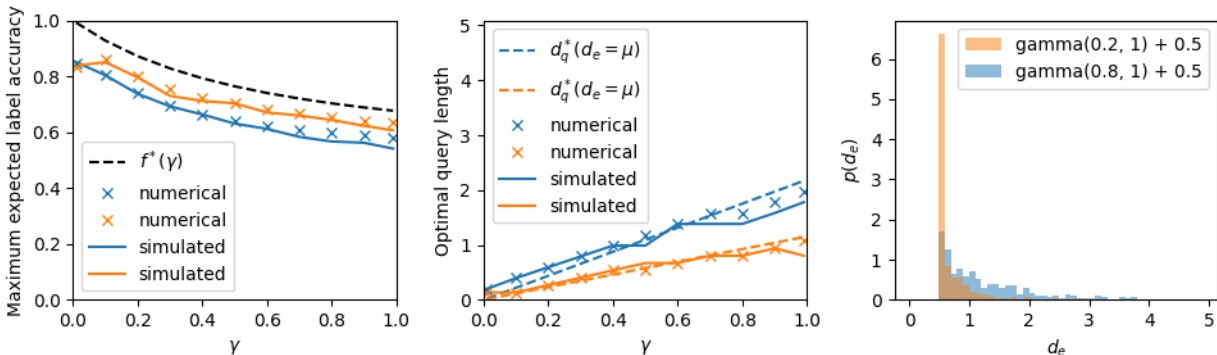

Figure 9: We validate the theory for stochastic event lengths drawn from two gamma distributions with different shape parameters, but the same scale parameter. We show the expected label accuracy (left panel), the optimal query length (middle panel), and the considered event length distributions (right panel).

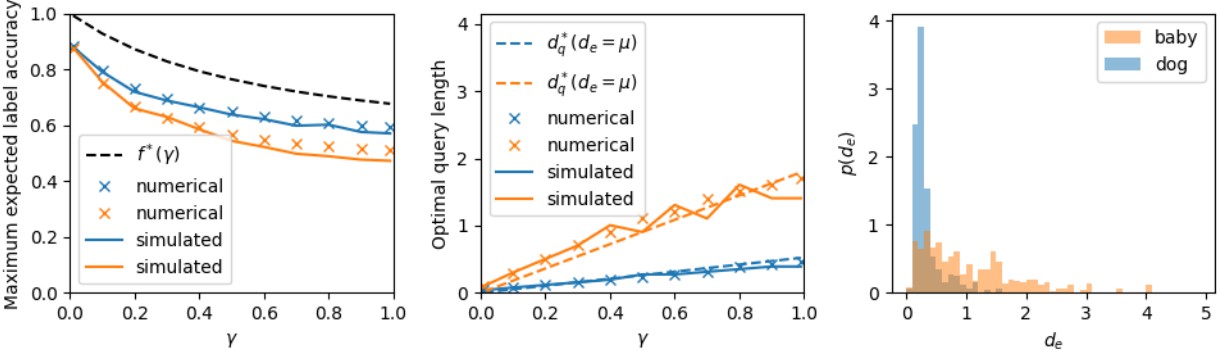

Figure 10: Barking dog and crying baby event length distributions from the NIGENS dataset (Trowitzsch et al., 2019). These annotations have been made with a strong guarantee for high quality onsets and offsets.

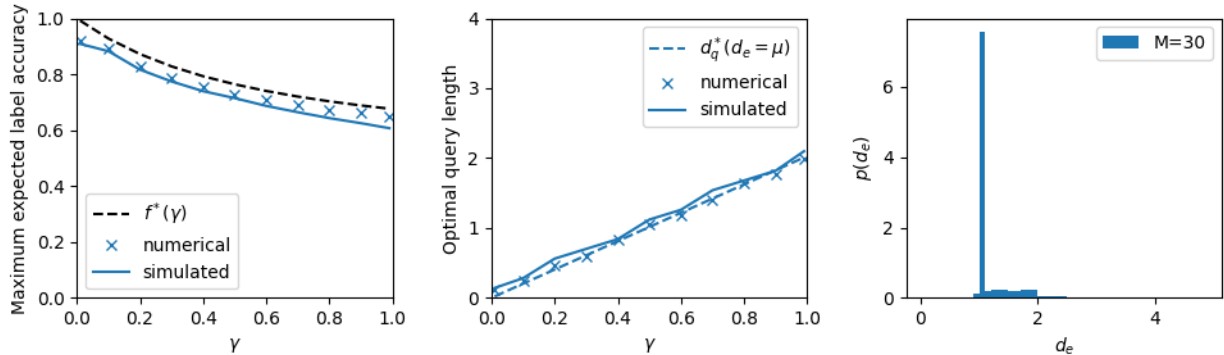

Figure 11: We validate the theory for multiple events of length $d_e = 1$. We show the expected label accuracy (left panel), the optimal query length (middle panel), and the considered event length distributions (right panel). Note that presence events longer than 1 can occur if two or more events overlap. We sample 30 events with event length $d_e = 1$ occur at random for each audio recording in this simulation.

### 6.1.3 Multiple Events with Stochastic Length

In these simulations we allow multiple events to occur in the same recording ($M > 1$). In Figure 11 we show the results of sampling 30 events of length $d_e = 1$ for each audio recording. This does have an effect on the expected maximum label accuracy and the corresponding query length, though the impact is relatively modest. In Figure 12 we show the results of sampling 50 events of length $d_e = 1$ for each audio recording. This is an extreme case, where the event density of the recording is very high. These results demonstrate that even under high event densities, the simulated maximum accuracy follows the theoretical predictions closely. This confirms that the single-event theory provides a robust upper bound even when assumptions about event sparsity are strongly violated.

### 6.2 Annotation Cost for Maximum Expected Label Accuracy given Overlap

Achieving maximum expected label accuracy comes at a cost, and understanding this cost trade-off is essential for practical annotation efforts. The cost model we employ accounts for both the time spent listening to audio and the effort required to label presence or absence events.

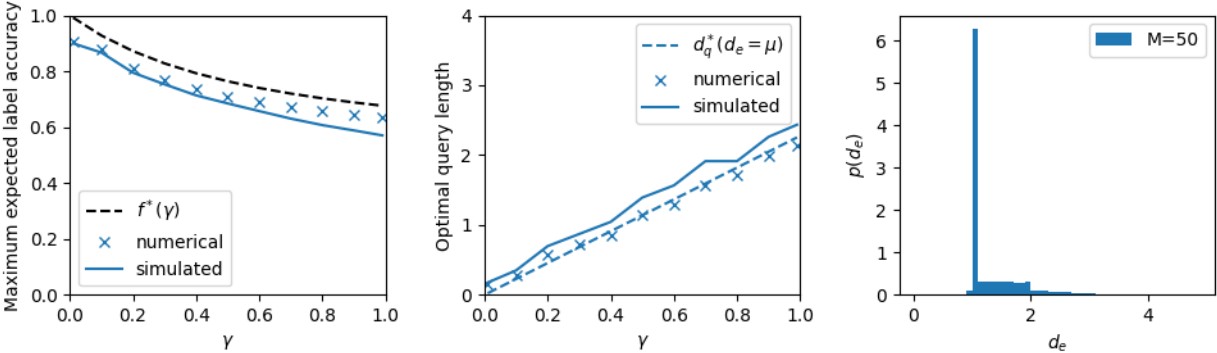

Figure 12: We validate the theory for multiple events of length $d_e = 1$. We show the expected label accuracy (left panel), the optimal query length (middle panel), and the considered event length distributions (right panel). Note that presence events longer than 1 can occur if two or more events overlap. We sample 50 events with event length $d_e = 1$ occur at random for each audio recording in this simulation.

### 6.2.1 Formalizing the Cost Model

The derived theory for the optimal query length allows us to analyze the cost of achieving maximum expected label accuracy under different annotator models for FIX weak labeling. We assume that the whole audio recording of length $T$ is listened to. The key difference in cost between the FIX and ORC weak labeling method is the number of segments ($B$) that need to be given a presence or absence label. We formalize a cost model as:

$$C(T, B) = (1 - r)T + rB, \tag{19}$$

where $1 - r$ represents the cost of listening to one second of audio (cost per second), and $r$ represents the cost of answering a query (cost per query). The term $(1 - r)T$ therefore represents the cost of listening to $T$ seconds of audio, and the term $rB$ the cost of assigning $B$ presence or absence labels. Using this cost model, we calculate the cost of annotating an audio recording of length $T$ with $M$ sound events of length $d_e = 1$ using either FIX or ORC weak labeling. For FIX, the number of queries that maximize expected label accuracy is given by $B_{\text{FIX}}^* = T/d_q^*$ (see Theorem 4). For ORC, achieving an expected label accuracy of 1 requires at least $B_{\text{ORC}}^* = 2M + 1$ queries.

In practice, we do not know the number of events $M$. To explore potential overestimation of $M$ when, for example, using a weak labeling process that tries to mimic ORC weak labeling, we model $B_{\text{ORC}}$ as a multiple of the necessary number of queries: $B_{\text{ORC}} = sB_{\text{ORC}}^*$, where $s \in \{1, 2, 4, 8\}$ represents the degree of overestimation. This approach captures scenarios where the number of events are either precisely estimated ($s = 1$) or significantly overestimated ($s = 8$) during the annotation process. In practice, $B_{\text{ORC}}$ could be set based on a bound on $M$. For example, by estimating a maximum expected number of sound events in a recording, $M_{\text{max}}$, based on knowledge of typical event density, or characteristics of the audio recording. We assume that overestimation by more than a factor of 8 is unlikely. The relative cost between FIX and ORC weak labeling can then be computed as:

$$\frac{C_{\text{FIX}}}{C_{\text{ORC}}} = \frac{C(T, B_{\text{FIX}}^*)}{C(T, B_{\text{ORC}})}, \tag{20}$$

where a ratio larger than 1 indicates that FIX is more costly than ORC, and a ratio smaller than 1 indicates that FIX is less costly than ORC.

### 6.2.2 Effect of annotator criteria ($\gamma$) and cost ratio ($r$).

Figure 13 (left) shows the relative cost for varying annotator criteria $\gamma \in [0.1, 1]$. As $\gamma \to 0.1$, the cost of FIX increases sharply, reflecting the need for an infinitely large number of queries to achieve an expected label accuracy of 1. In practice, achieving perfect accuracy with FIX is infeasible due to the associated cost.

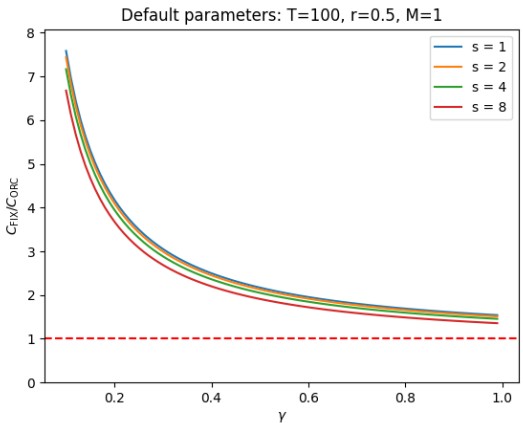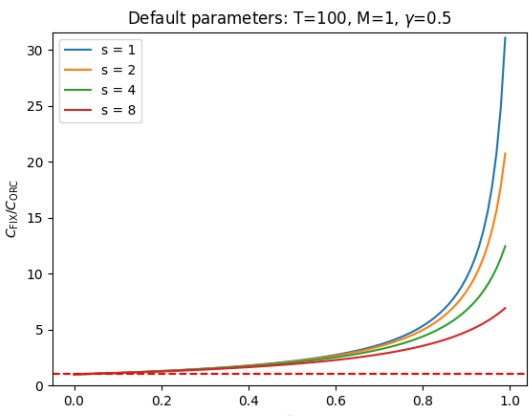

Figure 13: The relative cost of FIX and ORC for varying annotator criteria $\gamma$ (left), and cost ratios $r$ (right). The default parameters are: $T = 100$, $r = 0.5$, $M = 1$ and $\gamma = 0.5$. We simulate overestimating the number of needed queries $B_{\text{ORC}} = s(2M + 1)$ by a factor of $s$ for $s \in \{1, 2, 4, 8\}$ to see how this affects the relative cost. The cost of FIX is greater than the cost of ORC above the dashed red line where the cost ratio is 1.

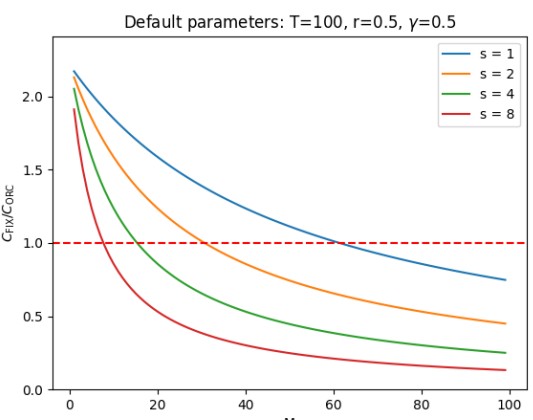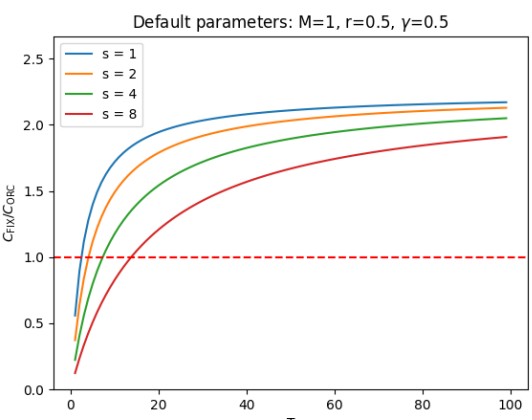

Figure 14: The relative cost of FIX and ORC for varying number of sound events $M$ (left) and recording lengths $T$ (right). The default parameters are: $T = 100$, $r = 0.5$, $M = 1$ and $\gamma = 0.5$. We simulate overestimating the number of needed queries $B_{\text{ORC}} = s(2M + 1)$ by a factor of $s$ for $s \in \{1, 2, 4, 8\}$ to see how this affects the relative cost. The cost of FIX is greater than the cost of ORC above the dashed red line where the cost ratio is 1.

For higher $\gamma$, the cost of FIX becomes more comparable to ORC. However, combining this with Theorem 3 reveals that FIX can either match ORC in cost but with lower expected accuracy or achieve similar accuracy at a much higher cost.

The right panel of Figure 13 examines the impact of the cost ratio $r$. Across all tested values, ORC remains less costly than FIX in the default setting ($T = 100$, $r = 0.5$, $\gamma = 0.5$, $M = 1$). This confirms that the relative cost advantage of ORC is robust to changes in $r$.

### 6.2.3 Effect of number of events ($M$) and recording length ($T$).

Figure 14 explores the impact of $M$ and $T$ on the relative cost. In the left panel, we see that for $s = 1$, ORC is less costly than FIX when the number of events is below 60. However, as $s$ increases to 8, FIX becomes

less costly when at most 10 events are present. These results indicate that the relative cost depends heavily on the density of sound events in the recording and the estimated annotation budget for ORC.

In the right panel, varying $T$ shows a similar trend. For shorter recordings (high event density), ORC loses its cost advantage. However, it's important to note that the maximum achievable expected label accuracy with FIX under default settings ($\gamma = 0.5$) is $f^*(0.5) \approx 0.76$, whereas ORC achieves 1.0. In such cases, the additional cost of ORC may be justified by the significantly higher label quality.

While these results indicate that the relative cost depends on the sound event density, we should remember that we are considering weak labeling of presence events. This implies that all $M$ events in this analysis are treated as non-overlapping, as the annotation task does not consider temporal overlaps for this analysis. The scenario of $M > 60$ non-overlapping events of length 1 in a recording of length $T = 100$ is therefore unlikely in practice. Similarly, estimating 10 events as 80 (modeled by $s = 8$) for an audio recording of length $T = 100$ represents a substantial overestimation and seems improbable given the capabilities of modern sound event detection tools.

## 7 Related Work

This work introduces a framework for characterizing segmentation label noise in FIX weak labeling, a largely unexplored area. Below, we review studies addressing noisy labels and approaches to mitigate their effects, with a focus on weak labeling in audio and related domains.

### 7.1 Understanding Noisy Labels

Noisy labels are a partial description of the target model, influencing its performance. Early work by Liang et al. (2009) introduced the concept of *measurements* for conditional exponential families, encompassing labels and constraints for model learning with minimal human input—a goal shared by this work.

In deep learning, the ability of models to overfit noisy labels has prompted studies into the relationship between noise rate and generalization (Zhang et al., 2021; Chen et al., 2019). Research on class label noise often assumes a noise transition matrix (Li et al., 2021) but rarely considers spatially correlated errors like those arising in segmentation tasks (Yao et al., 2023). For audio, Hershey et al. (2021) demonstrated that training on strongly labeled data yields better results than weakly labeled data, highlighting the need for precise labels, particularly in evaluation. In multi-modal tasks, such as audio-visual video parsing, a key challenge is *modality-specific label noise*, where a video-level tag may apply to the audio stream but not the visual, or vice versa (Cheng et al., 2022; Zhou et al., 2024). Our work focuses specifically on characterizing the *segment label noise* that arises from the temporal misalignment between fixed-length query segments and true event boundaries.

### 7.2 Mitigating Noisy Labels

Several strategies address noisy labels, including regularization techniques like dropout (Srivastava et al., 2014), data augmentation (Shorten & Khoshgoftaar, 2019), and specialized loss functions (Fonseca et al., 2019). For weakly labeled audio, Dinkel et al. (2022) proposed a pseudo-labeling approach, iteratively refining labels to improve training performance. Despite these advances, most methods focus on training labels and offer limited insights into noisy evaluation labels, underscoring the need for frameworks that quantify label noise, such as the one proposed in this work.

### 7.3 Strong vs. Weak Labeling

Strong labeling, where the annotator provides the event boundaries and the class label, while often precise, is resource-intensive and subject to annotator variability (Mesaros et al., 2017). In bioacoustics, experts use spectrograms for efficient annotation (Cartwright et al., 2017), but the reliance on specialists limits scalability. Weak labeling, by contrast, simplifies the annotation task, which is especially important for crowd-sourced annotations, enabling broader data collection (Martin-Morato & Mesaros, 2023). However, segment label noise, especially at event boundaries, remains a significant challenge.

| Dataset | Task | Fixed Length |
|---|---|---|
| CHIME (Foster et al., 2015a) | Single-pass multi-label | 4 seconds |
| AudioSet (Gemmeke et al., 2017b) | Single-pass multi-label | 10 seconds |
| MAESTRO Real (Martin-Morato & Mesaros, 2023) | Single-pass multi-label | 10 seconds |
| OpenMIC-2018 (Humphrey et al., 2018b) | Multi-pass binary-label | 10 seconds |

Table 1: Large-scale audio datasets using variations of FIX weak labeling.

Large-scale audio datasets employing FIX weak labeling are summarized in Table 1. Two common annotation tasks are single-pass multi-label and multi-pass binary-label annotation (Cartwright et al., 2019). Single-pass multi-label annotation asks annotators to recognize the presence of multiple event classes during a single pass through the data. In contrast, multi-pass binary-label annotation asks annotators to detect the presence or absence of a single event class at a time through multiple passes through the data.

Cartwright et al. (2019) studied the trade-offs between these tasks and found that binary labeling is preferable when high recall is required. For example, AudioSet (Gemmeke et al., 2017b) employs single-pass multi-label annotation with non-overlapping 10-second segments, which limits temporal resolution. Conversely, MAESTRO Real (Martin-Morato & Mesaros, 2023) uses overlapping 10-second segments with a 9-second overlap, increasing the accuracy of the derived labels.

The choice of segment length and overlap significantly impacts the utility of weak labeling. For example, while overlapping segments increase label accuracy (Martin-Morato & Mesaros, 2023), they still fail to distinguish events occurring close in time. Current work aims to better understand the effect of different choices of the segment length for FIX weak labeling.

### 7.4 Contributions of This Work

Existing research focuses predominantly on class label noise or assumes noise independence. This work extends these efforts by characterizing segment label noise specific to FIX weak labeling, providing a foundation for improving both training and evaluation processes in weakly labeled datasets.

## 8 Discussion

FIX labeling has been employed in many works, with varying degrees of complexity. Theorem 2 provides a useful rule of thumb for selecting the best segmentation length for a given event length, and Eq. 13 provides a way to use this theorem to analyze stochastic event length distributions. Our results suggest that, in most cases, knowing the average event length provides a good estimate, but understanding the (approximate) distribution of event lengths improves the analysis.

**Implications for Practical Annotation**

Our theoretical analysis positions FIX weak labeling as a baseline strategy: it is simple and scalable but inherently limited by segment label noise. By quantifying this limitation, our work provides the necessary theoretical justification for moving towards adaptive methods that aim to approximate the oracle process.

The analysis highlights the trade-offs in label accuracy and annotation cost between FIX and ORC weak labeling. While FIX can be less costly under specific conditions (e.g., high event density), these conditions are unlikely to occur in real-world annotation tasks. Furthermore, even in cases where FIX is less costly, its significantly lower label accuracy ($f^*(0.5) \approx 0.76$ vs. $1.0$ for ORC) can negate its cost advantage. This gap represents the "accuracy cost of weakness" that is inherent to any non-adaptive weak labeling strategy. This gap only increases when $d_q$ is chosen sub-optimally, which is often the case in practice due to budget considerations. Given the rarity of extreme event densities and the importance of high-quality labels, ORC is likely the better theoretical choice for most annotation tasks.

However, ORC weak labeling is not available in practice since it uses the true event boundaries. This provides a clear theoretical justification for developing adaptive weak labeling methods, which aim to approximate

the ORC process. For instance, methods that use active learning or change-point detection to define query boundaries (Martinsson et al., 2024; Kim et al., 2023) are practical attempts to bridge the gap between FIX and ORC. Martinsson et al. (2024) empirically evaluates an adaptive change-point detection method (A-CPD) and compares that to FIX weak labeling and ORC weak labeling, showing the benefit of an adaptive weak labeling method for annotation of sound events. Our work provides the tools to quantify the maximum potential accuracy gain for such methods over a simple FIX baseline, offering a principled way to evaluate the trade-off between the complexity of an adaptive strategy and its achievable accuracy. Future research should focus on mitigating the potential biases when modeling ORC weak labeling (e.g., annotation errors, overfitting to sparse events) while retaining its theoretical advantages.

**Implications for Model Evaluation**

Despite the extensive focus on noisy training labels, evaluation labels are often implicitly assumed to be perfect. As emphasized in the introduction, inaccurate evaluation labels present a significant challenge. When noise is present in both training and evaluation data, we risk selecting models that merely replicate the evaluation noise, potentially overlooking those with superior generalization abilities. This echoes the central concern highlighted by Görnitz et al. (2014). We can use Theorem 3 to understand the properties of the best performing model when the evaluation data contains FIX weak labels. For example, for $\gamma = 0.5$ the annotations will at most have an expected label accuracy of $f^*(0.5) \approx 0.76$. The "best" performing model will therefore be a model that mimics this specific noise profile. Our theory thus provides a better understanding of the target that models are optimizing for when evaluated on weakly labeled data.

This is also relevant for standard sound event detection (SED) evaluation metrics, such as the segment-based $F_1$ score (Mesaros et al., 2016), which divide audio into fixed-length segments. When using ground truth labels for evaluation, we effectively have an annotator with $\gamma \to 0$. The expected label accuracy then becomes $f(d_q) = d_e/(d_e + d_q)$, where $d_q$ is the segment length. This formula shows that a small $d_q$ minimizes segment label noise, but choosing a very small segment length negates the desired effect of mitigating temporal imprecision in the ground truth and also increases computational cost. The theory presented here can help inform such trade-offs.

**Theoretical Properties and Validation**

The expression for expected label accuracy derived in this paper applies to the simplest scenario, where only a single event with deterministic length is present. In all of our results, we observe that $f^*(\gamma)$ is greater than or equal to the expected and average label accuracy that FIX weak labeling achieve for more complex distributions. This suggests that $f^*(\gamma)$ can be considered an upper bound for a given annotation process. However, a formal proof showing that adding more events or introducing event length variability leads to a harder distribution to annotate is beyond the scope of this paper.

To connect this theoretical framework to a real-world setting, we conducted an empirical analysis using the weakly and strongly labeled versions of the AudioSet dataset, as detailed in Appendix A.5. By treating the 10-second weak labels of AudioSet as the output of a FIX process, we calculated the empirical label accuracy against the corresponding strong labels. Our theoretical model, when applied to the event length distribution of the "Animal" class, accurately predicted this empirical accuracy for a presence criterion of $\gamma \approx 0.26$. This serves as an empirical validation of our framework on a large-scale dataset.

**Generalization and Future Directions**

While our work is grounded in audio event detection, the core principles are broadly generalizable because the mathematics depends only on two fundamental quantities: the event duration $d_e$, and the query segment length $d_q$. These quantities can be directly mapped to other domains, e.g., video action spotting, electrocardiography, seismology, or high-frequency trading. Consequently, our core results (Theorems 1-3) transfer directly to these domains, provided three conditions hold: (i) the events uniformly distributed locally in time, making the uniform relative offset a reasonable model (as empirically verified in Appendix A.4), (ii) the annotation process relies on observing a minimum fraction $\gamma$ of an event, and (iii) the events are suf-

ficiently sparse. This framework can also be extended to higher dimensions, such as analyzing the weak labeling of rectangles in images or cubes in point clouds.

Finally, if the same presence criterion $\gamma$ is applicable for all event classes, Theorem 1 applies to the joint event length distribution. However, real-world presence criteria for different event classes may vary, requiring more complex models. Future empirical studies on annotator behavior could help refine this model and improve its practical applicability.

## 9 Conclusions

This study introduces a novel theoretical framework for understanding the trade-offs between label accuracy and annotation cost in weak labeling methods, particularly focusing on sound event detection where weak labeling is often employed to reduce annotation costs. We specifically compared fixed-length (FIX) and oracle (ORC) approaches.

We have demonstrated that FIX weak labeling, while cost-effective in specific scenarios, is inherently limited by segment label noise. The expressions we derived theoretically provide actionable insights into optimizing segment length for maximizing expected label accuracy under FIX. However, these results also underscore the fundamental trade-offs: shorter segments improve alignment with event boundaries but significantly increase annotation cost, while longer segments reduce cost at the expense of accuracy. In addition, how short these segments can be chosen depends on the ability of the annotator to detect presence of fractions of the events. In contrast, ORC labeling achieves perfect accuracy but can incur higher costs if events are very dense and the number of events are overestimated.

Our findings have several practical implications:

- **Annotation Strategy:** FIX weak labeling remains a robust, scalable choice for many practical applications. However, when high label accuracy is essential, ORC weak labeling—or adaptive methods approximating it—should be prioritized.

- **Adaptive Techniques:** Theoretical justification for adaptive weak labeling methods, e.g., methods based on active learning or iterative refinement, that mimic ORC weak labeling, which suggests promising avenues for improving annotation efficiency without compromising accuracy.

- **Evaluation Criteria:** Our analysis highlights the potential biases introduced by segment-level label noise in evaluating sound event detection models. Therefore, carefully aligning evaluation criteria with the intended model properties is critical.

Future research should address several limitations and extensions identified in our study. Developing practical approaches that reliably mimic ORC weak labeling by estimating the query segments without introducing a lot of unwanted bias in the labels remains an open challenge. Additionally, extending this framework to multi-dimensional data and multiple presence classes could broaden its applicability to other domains, such as medical imaging and point clouds.

In conclusion, the insights presented in this work offer a foundation for optimizing weak labeling processes, balancing cost and accuracy to meet the needs of diverse machine learning applications. By refining annotation strategies and leveraging adaptive methods, researchers can enhance the quality of labeled datasets. This, in turn, will drive advancements in supervised learning across domains, building upon the foundational understanding presented in this work.

### Acknowledgments

The authors would like to express their gratitude to the following individuals and organizations for their contributions and support:

- **Annamaria Mesaros**: Provided insightful questions and feedback on an early draft of this work.

- **Magnus Oskarsson**: Contributed a proof sketch for Theorem 2 and gave feedback on an early draft.

- **Edvin Listo Zec**: Assisted with the proof of Theorem 2.

- This work was supported by the Swedish Foundation for Strategic Research (SSF; FID20-0028) and Sweden's Innovation Agency (2023-01486).

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

# A    Appendix

We do not include all simplifications of expressions in the proofs, but we do provide the code for a symbolic mathematics solver (SymPy) at GitHub[2], where all results can be verified. The notebook named "symbolic_verification_of_analysis.ipynb" can be used to verify the analysis.

## A.1    Proof of Theorem 1

We will derive an expression for the expected query segment accuracy given overlap with a single event in terms of $d_e$, $d_q$, and $\gamma$, under all possible assumptions which will prove Theorem 1.

*Proof.* We need to consider two main assumptions. The first assumption is that the presence criterion for the annotator can be fulfilled, that is, $d_q \geq \gamma d_e$, and the second assumption is that the annotator presence criterion can not be fulfilled, that is, $d_q < \gamma d_e$. This happens if the query segment length is so short that it can never cover a large enough fraction of the event of interest to make presence detection feasible.

**Assumption 1.** The annotator presence criterion can be fulfilled ($d_q \geq \gamma d_e$).

Under this assumption there are two possible cases for the relation between $d_q$ and $d_e$, either the event length is longer or equal to the query segment length, $d_e \geq d_q$ (case i), or the event length is shorter than the query segment length, $d_e < d_q$ (case ii). In Figure 15, we plot the query segment accuracy, $F(e_t, q, \gamma)$, for $t \in [0, d_e + d_q]$ for case (i) on the left, and case (ii) on the right. We describe in more detail in Appendix A.1.1 how the query segment accuracy behaves as a function of different amounts of overlap between the query segment and the event. Briefly, what we see in Figure 15 is that initially there is arbitrarily little overlap ($t_0^{(i)}$ and $t_0^{(ii)}$), an absence label is given to the query segment and the accuracy is therefore 1. Then the accuracy decrease linearly with the amount of overlap until the presence criterion is fulfilled and a presence label is given ($t_1^{(i)}$ and $t_1^{(ii)}$). After that, the accuracy linearly increase with the amount of overlap between the event and query segment until we reach a ceiling for the accuracy when either the whole query segment is inside the event ($t_2^{(i)}$) or the query segment covers the whole event ($t_2^{(ii)}$). Finally, the overlap between the query segment and the event starts to decrease again ($t_3^{(i)}$ and $t_3^{(ii)}$), and everything is symmetrical.

We continue by dropping the case superscripts show in the figure for $A_1, \ldots, A_3$ and $t_0, \ldots, t_5$, and only provide the full proof for case (i), but the proof for case (ii) is similar. In both cases the area $A$ in Eq. 8 can be divided into five distinct parts:

$$A = 2A_1 + 2A_2 + A_3, \tag{21}$$

where $A_1$ and $A_2$ are counted twice due to symmetry.

---

[2]`https://github.com/johnmartinsson/the-accuracy-cost-of-weakness`

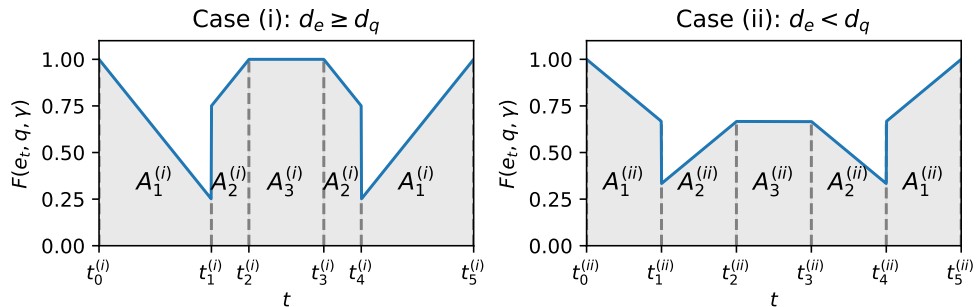

Figure 15: Assuming $d_q \geq d_e \gamma$, we plot the query segment accuracy, $F(e_t, q, \gamma)$, for $t \in [0, d_e + d_q]$, where $t_0 = 0$ and $t_5 = d_e + d_q$. Case (i) where $d_e \geq d_q$ is shown in the left panel, and case (ii) where $d_e < d_q$ is shown in the right panel.

The variables $t_0, t_1, \ldots, t_5$, represent the different states $t$ of overlap where the discontinuities of $F(e_t, q, \gamma)$ occur, and using these we can express the areas as the following integrals:

$$A_1 = \int_{t_0}^{t_1} F(e_t, q, \gamma) \mathrm{d}t = \int_{t_4}^{t_5} F(e_t, q, \gamma) \mathrm{d}t, \tag{22}$$

and

$$A_2 = \int_{t_1}^{t_2} F(e_t, q, \gamma) \mathrm{d}t = \int_{t_3}^{t_4} F(e_t, q, \gamma) \mathrm{d}t, \tag{23}$$

due to symmetry, and

$$A_3 = \int_{t_2}^{t_3} F(e_q, q, \gamma) \mathrm{d}t. \tag{24}$$

We use that the query segment accuracy $F(e_t, q, \gamma)$ is linear in each interval, which means that the areas can be expressed as

$$A_1 = \frac{F(e_{t_0}, q, \gamma) + F(e_{t_{1-}}, q, \gamma)}{2}(t_1 - t_0), \tag{25}$$

$$A_2 = \frac{F(e_{t_{1+}}, q, \gamma) + F(e_{t_2}, q, \gamma)}{2}(t_2 - t_1), \tag{26}$$

and

$$A_3 = \frac{F(e_{t_2}, q, \gamma) + F(e_{t_3}, q, \gamma)}{2}(t_3 - t_2), \tag{27}$$

where $t^-$ indicate that we approach the discontinuity at $t$ from below and $t^+$ from above. We now only need to express $t_0, \ldots, t_3$ and $F(e_{t_0}, q, \gamma), \ldots, F(e_{t_3}, q, \gamma)$ in terms of $d_e$, $d_q$ and $\gamma$ to conclude the proof. For brevity, these have been provided in Table 2. See section A.1.1 for details on how to express these in terms of $d_q$, $d_e$ and $\gamma$.

We provide the steps for case (i), and leave the derivation for case (ii) to the reader. We substitute the expressions for case (i), provided in Table 2, into equations Eq. 25-27, and the resulting expressions for the areas $A_1^{(i)}$, $A_2^{(i)}$, and $A_3^{(i)}$ into Eq. 21 which give

| Case (i), $d_e \geq d_q$ | | Case (ii), $d_e < d_q$ | |
|---|---|---|---|
| $t_0^{(i)} = 0$ | $F(e_{t_0}^{(i)}, q, \gamma) = 1$ | $t_0^{(ii)} = 0$ | $F(e_{t_0}^{(ii)}, q, \gamma) = 1$ |
| $t_1^{(i)} = \gamma d_e$ | $F(e_{t_1^-}^{(i)}, q, \gamma) = \frac{d_q - \gamma d_e}{d_q}$ | $t_1^{(ii)} = \gamma d_e$ | $F(e_{t_1^-}^{(ii)}, q, \gamma) = \frac{d_q - \gamma d_e}{d_q}$ |
| $t_2^{(i)} = d_q$ | $F(e_{t_1^+}^{(i)}, q, \gamma) = \frac{\gamma d_e}{d_q}$ | $t_2^{(ii)} = d_e$ | $F(e_{t_1^+}^{(ii)}, q, \gamma) = \frac{\gamma d_e}{d_q}$ |
| $t_3^{(i)} = d_e$ | $F(e_{t_2}^{(i)}, q, \gamma) = 1$ | $t_3^{(ii)} = d_q$ | $F(e_{t_2}^{(ii)}, q, \gamma) = \frac{d_e}{d_q}$ |
| | $F(e_{t_3}^{(i)}, q, \gamma) = 1$ | | $F(e_{t_3}^{(ii)}, q, \gamma) = \frac{d_e}{d_q}$ |

Table 2: A summary of the derived expressions for $t_0, \ldots, t_3$ and $F(e_{t_0}, q, \gamma), \ldots, F(e_{t_3}, q, \gamma)$ for each case. $F(e_{t_1^-}, q, \gamma)$ and $F(e_{t_1^+}, q, \gamma)$ denotes the limits when approaching $t_1$ from below and above respectively.

$$
\begin{aligned}
A^{(i)} &= \frac{2}{2}(1 + \frac{d_q - \gamma d_e}{d_q})\gamma d_e + \frac{2}{2}(1 + \frac{\gamma d_e}{d_q})(d_q - \gamma d_e) + (d_e - d_q) \\
&= (2d_q - \gamma d_e)\frac{\gamma d_e}{d_q} + (d_q + \gamma d_e)(d_q - \gamma d_e)\frac{1}{d_q} + (d_e - d_q) \\
&= \frac{1}{d_q}(2\gamma d_q d_e - \gamma^2 d_e^2 + \cancel{d_q^2} - \gamma^2 d_e^2 + d_e d_q - \cancel{d_q^2}) \\
&= \frac{1}{d_q}(2\gamma d_q d_e - 2\gamma^2 d_e^2 + d_e d_q) \\
&= \frac{d_e}{d_q}(2\gamma d_q - 2\gamma^2 d_e + d_q).
\end{aligned}
$$

Finally, by substituting $A$ for $A^{(i)}$ in Eq. 8 we arrive at

$$
\frac{A^{(i)}}{d_e + d_q} = \frac{d_e(2\gamma d_q - 2\gamma^2 d_e + d_q)}{d_q(d_e + d_q)} \tag{28}
$$

which shows that Eq. 9 holds for case (i) under the assumption that $d_q \geq \gamma d_e$. Similarly, this also holds for case (ii).

**Assumption 2.** The annotator presence criterion can not be fulfilled ($d_q < \gamma d_e$).

When the presence criterion can not be fulfilled we never get any presence labels, this means that the fraction of the query segment that overlaps with an event is always incorrectly given an absence label. When the query segment completely overlaps with an event the query segment accuracy will be 0 (seen between $t_1$ and $t_2$ in Figure 16).

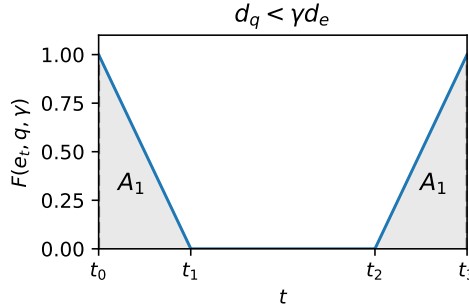

Figure 16: Assuming that $d_q < \gamma d_e$, we plot the query segment accuracy, $F(e_t, q, \gamma)$, for $t \in [0, d_e + d_q]$, where $t_0 = 0$ and $t_3 = d_e + d_q$.

The area $A_1$ is counted twice due to symmetry. The discontinuity at $t_1$ occurs for the smallest $t \in [0, d_e + d_q]$ for which $F(e_t, q, \gamma) = 0$, which happens for the smallest $t$ for which the whole query segment overlaps with the event $|e \cap q| = d_q$ at $t = d_q$. We therefore have that $t_1 - t_0 = t_3 - t_2 = d_q$.

When there is no overlap between the query segment and the event giving a presence label is always correct, thus $F(e_{t_0}, q, \gamma) = 1$. However, giving an absence label to a query segment that completely overlaps with an event gives the query segment accuracy 0, thus $F(e_{t_1}, q, \gamma) = 0$. The total area under the curve is therefore $2A_1 = d_q$ and by normalizing with $t_3 - t_0 = d_e + d_q$, we get $d_q/(d_e + d_q)$, which proves the $d_q < \gamma d_e$ case of Eq. 9, and concludes the proof.

$\square$

### A.1.1 Details on the expressions in Table 2

This section provides a detailed explanation of the values presented in Table 2. For each case (i) and (ii), we will define the specific time points $t_0, t_1, t_2, t_3$ where the query segment accuracy function $F(e_t, q, \gamma)$ changes, and explain the corresponding value of the function at these points based on the overlap between the event $e_t$ and the query segment $q$. The states $t_4$ and $t_5$ are analogous to $t_1$ and $t_0$, respectively, and therefore not illustrated. The difference is that the amount of overlap between the query segment and event decreases (instead of increases) when approaching these states.

**Case (i):** $d_e \geq d_q$

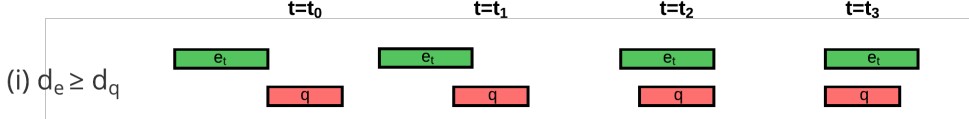

Figure 17: An illustration of how the sound event $e_t$ and the query segment $q$ overlap at the four distinct states $t = t_0, \ldots, t_3$ for case (i) where $d_e \geq d_q$.

- $t_0^{(i)}$: At $t_0^{(i)} = 0$, the end of the event $e_t$ aligns perfectly with the beginning of the query segment $q$. This means there is no overlap between the event and the query segment ($|e_{t_0^{(i)}} \cap q| = 0$). Therefore, assuming the annotator absence criterion applies, the query segment accuracy is $F(e_{t_0^{(i)}}, q, \gamma) = \frac{d_q - |e_{t_0^{(i)}} \cap q|}{d_q} = \frac{d_q - 0}{d_q} = 1$.

- $t_1^{(i)}$: The time $t_1^{(i)} = \gamma d_e$ represents the point where the annotator presence criterion is first met. Before this point ($t < t_1^{(i)}$), the overlap $|e_t \cap q|$ is less than $\gamma d_e$, and the query segment accuracy is given by $F(e_t, q, \gamma) = \frac{d_q - |e_t \cap q|}{d_q}$. As $t$ approaches $t_1^{(i)}$ from the left, $|e_t \cap q|$ approaches $\gamma d_e$, hence $\lim_{t \to t_1^-} F(e_t, q, \gamma) = \frac{d_q - \gamma d_e}{d_q}$. At $t = t_1^{(i)}$, the presence criterion is met, and the accuracy function switches to $F(e_t, q, \gamma) = \frac{|e_t \cap q|}{d_q}$. As $t$ approaches $t_1^{(i)}$ from the right, $|e_t \cap q|$ is slightly greater than $\gamma d_e$, and $\lim_{t \to t_1^+} F(e_t, q, \gamma) = \frac{\gamma d_e}{d_q}$. This transition is visually represented in Figure 17 at time $t = t_1$.

- $t_2^{(i)}$: At $t_2^{(i)} = d_q$, the entire query segment $q$ is fully contained within the event $e_t$. This means the overlap is maximal: $|e_{t_2^{(i)}} \cap q| = d_q$. Since the presence criterion is met, the query segment accuracy is $F(e_{t_2^{(i)}}, q, \gamma) = \frac{|e_{t_2^{(i)}} \cap q|}{d_q} = \frac{d_q}{d_q} = 1$. This behavior is visually represented in Figure 17 at time $t = t_2$, where the green box representing the event fully covers the red box representing the query segment.

- $t_3^{(i)}$: At $t_3^{(i)} = d_e$, the entire query segment $q$ still fully overlaps with the event $e_t$. Similar to $t_2$, the overlap is $|e_{t_3^{(i)}} \cap q| = d_q$, and therefore $F(e_{t_3^{(i)}}, q, \gamma) = \frac{|e_{t_3^{(i)}} \cap q|}{d_q} = \frac{d_q}{d_q} = 1$. This is depicted in Figure 17 at time $t = t_3$.

**Case (ii):** $d_e < d_q$

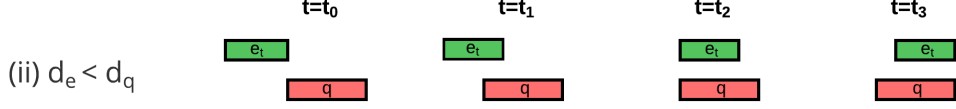

Figure 18: An illustration of how the sound event $e_t$ and the query segment $q$ overlap at the four distinct states $t = t_0, \ldots, t_3$ for case (ii) where $d_e < d_q$.

- $t_0^{(ii)}$: At $t_0^{(ii)} = 0$, the end of the event $e_t$ aligns perfectly with the beginning of the query segment $q$. There is no overlap ($|e_{t_0^{(ii)}} \cap q| = 0$). Assuming the annotator absence criterion applies, the query segment accuracy is $F(e_{t_0^{(ii)}}, q, \gamma) = \frac{d_q - |e_{t_0^{(ii)}} \cap q|}{d_q} = \frac{d_q - 0}{d_q} = 1$.

- $t_1^{(ii)}$: The time $t_1^{(ii)} = \gamma d_e$ again marks the point where the annotator presence criterion is first met. Before this ($t < t_1^{(ii)}$), the overlap $|e_t \cap q| < \gamma d_e$, and $F(e_t, q, \gamma) = \frac{d_q - |e_t \cap q|}{d_q}$. Approaching $t_1^{(ii)}$ from

the left, $|e_t \cap q| \to \gamma d_e$, thus $\lim_{t \to t_1^-} F(e_t, q, \gamma) = \frac{d_q - \gamma d_e}{d_q}$. At $t = t_1^{(ii)}$, the criterion is met, and the function becomes $F(e_t, q, \gamma) = \frac{|e_t \cap q|}{d_q}$. Approaching from the right, $|e_t \cap q|$ is slightly greater than $\gamma d_e$, so $\lim_{t \to t_1^+} F(e_t, q, \gamma) = \frac{\gamma d_e}{d_q}$. This transition is shown in Figure 18 at $t = t_1$.

- $t_2^{(ii)}$: At $t_2^{(ii)} = d_e$, the beginning of the event $e_t$ aligns with the beginning of the query segment $q$. At this point, the overlap is maximal, as the entire event is contained within the query segment: $|e_{t_2^{(ii)}} \cap q| = d_e$. Since the presence criterion is met, the query segment accuracy is $F(e_{t_2^{(ii)}}, q, \gamma) = \frac{|e_{t_2^{(ii)}} \cap q|}{d_q} = \frac{d_e}{d_q}$. This situation is illustrated in Figure 18 at $t = t_2$.

- $t_3^{(ii)}$: At $t_3^{(ii)} = d_q$, the end of the event $e_t$ aligns with the end of the query segment $q$. Similar to $t_2^{(ii)}$, the entire event is contained within the query segment, so the overlap is $|e_{t_3^{(ii)}} \cap q| = d_e$. Consequently, the query segment accuracy is $F(e_{t_3^{(ii)}}, q, \gamma) = \frac{|e_{t_3^{(ii)}} \cap q|}{d_q} = \frac{d_e}{d_q}$. This corresponds to the state depicted in Figure 18 at $t = t_3$.

Understanding these key time points and the corresponding query segment accuracy values is crucial for calculating the area under the curve, which represents the expected query segment accuracy.

## A.2 Proof of Theorem 2

*Proof.* We start by finding a unique critical point $d_q^*$ which makes $f'(d_q^*) = 0$ when $d_q \geq \gamma d_e$. We then show that $d_q^*$ is a global maximum by analyzing the boundaries of $f(d_q)$ on its' domain when $d_q \geq \gamma d_e$. We show that $f(d_q^*) \geq f(\gamma d_e)$ and that $f(d_q^*) \geq \lim_{d_q \to \infty} f(d_q)$. Since $d_q^*$ is a unique critical point we conclude that it must be a global maximum of the function $f(d_q)$ when $d_q \geq \gamma d_e$. Lastly, we show that $f(d_q^*) \geq f(\gamma d_e) \geq f(d_q)$ when $d_q < \gamma d_e$ which proves that $d_q^*$ is a global maximum of the function $f(d_q)$ for $d_q > 0$.

## 1. Finding the unique critical point $d_q^*$.

To find the critical points, we need to compute the derivative of $f(d_q)$ with respect to $d_q$ and set it to zero. Let $N(d_q) = d_e(-2d_e\gamma^2 + 2d_q\gamma + d_q)$ and $D(d_q) = d_q(d_e + d_q)$. Then $f(d_q) = \frac{N(d_q)}{D(d_q)}$. Using the quotient rule, the derivative is given by:

$$f'(d_q) = \frac{N'(d_q)D(d_q) - N(d_q)D'(d_q)}{[D(d_q)]^2}$$

First, we find the derivatives of the numerator and the denominator:

$$N'(d_q) = \frac{d}{dd_q}[d_e(-2d_e\gamma^2 + 2d_q\gamma + d_q)]$$
$$= d_e(0 + 2\gamma + 1)$$
$$= d_e(2\gamma + 1)$$

$$D(d_q) = d_q(d_e + d_q) = d_e d_q + d_q^2$$
$$D'(d_q) = \frac{d}{dd_q}[d_e d_q + d_q^2]$$
$$= d_e + 2d_q$$

Now, we plug these into the quotient rule formula:

$$f'(d_q) = \frac{[d_e(2\gamma + 1)][d_q(d_e + d_q)] - [d_e(-2d_e\gamma^2 + 2d_q\gamma + d_q)][d_e + 2d_q]}{[d_q(d_e + d_q)]^2}$$

To find the critical points, we set $f'(d_q) = 0$, which means the numerator must be zero:

$$[d_e(2\gamma + 1)][d_q(d_e + d_q)] - [d_e(-2d_e\gamma^2 + 2d_q\gamma + d_q)][d_e + 2d_q] = 0$$

Since $d_e > 0$, we can divide by $d_e$:

$$(2\gamma + 1)d_q(d_e + d_q) - (-2d_e\gamma^2 + 2d_q\gamma + d_q)(d_e + 2d_q) = 0$$

Expanding the terms:

$$(2\gamma + 1)(d_e d_q + d_q^2) - (-2d_e^2\gamma^2 - 4d_e d_q\gamma^2 + 2d_e d_q\gamma + 4d_q^2\gamma + d_e d_q + 2d_q^2) = 0$$
$$2\gamma d_e d_q + 2\gamma d_q^2 + d_e d_q + d_q^2 - (-2d_e^2\gamma^2 - 4d_e d_q\gamma^2 + 2d_e d_q\gamma + 4d_q^2\gamma + d_e d_q + 2d_q^2) = 0$$

Collecting and rearranging the terms to form a quadratic equation in $d_q$:

$$(2\gamma + 1 - 4\gamma - 2)d_q^2 + (2\gamma + 1 + 4\gamma^2 - 2\gamma - 1)d_e d_q + 2d_e^2\gamma^2 = 0$$
$$(-2\gamma - 1)d_q^2 + (4\gamma^2)d_e d_q + 2d_e^2\gamma^2 = 0$$
$$(2\gamma + 1)d_q^2 - 4\gamma^2 d_e d_q - 2d_e^2\gamma^2 = 0$$

Using the quadratic formula $d_q = \frac{-b \pm \sqrt{b^2 - 4ac}}{2a}$, where $a = 2\gamma + 1$, $b = -4d_e\gamma^2$, $c = -2d_e^2\gamma^2$:

$$d_q = \frac{4d_e\gamma^2 \pm \sqrt{(-4d_e\gamma^2)^2 - 4(2\gamma + 1)(-2d_e^2\gamma^2)}}{2(2\gamma + 1)}$$
$$= \frac{4d_e\gamma^2 \pm \sqrt{16d_e^2\gamma^4 + 8(2\gamma + 1)d_e^2\gamma^2}}{4\gamma + 2}$$
$$= \frac{4d_e\gamma^2 \pm \sqrt{16d_e^2\gamma^4 + 16d_e^2\gamma^3 + 8d_e^2\gamma^2}}{4\gamma + 2}$$
$$= \frac{4d_e\gamma^2 \pm \sqrt{8d_e^2\gamma^2(2\gamma^2 + 2\gamma + 1)}}{4\gamma + 2}$$
$$= \frac{4d_e\gamma^2 \pm 2d_e|\gamma|\sqrt{4\gamma^2 + 4\gamma + 2}}{2(2\gamma + 1)}$$

Since $\gamma > 0$, we have $|\gamma| = \gamma$:

$$d_q = \frac{4d_e\gamma^2 \pm 2d_e\gamma\sqrt{4\gamma^2 + 4\gamma + 2}}{2(2\gamma + 1)}$$
$$= \frac{2d_e\gamma^2 \pm d_e\gamma\sqrt{4\gamma^2 + 4\gamma + 2}}{(2\gamma + 1)}$$
$$= d_e\gamma\frac{2\gamma \pm \sqrt{4\gamma^2 + 4\gamma + 2}}{2\gamma + 1}$$

We note that $\sqrt{4\gamma^2 + 4\gamma + 2} = 2\sqrt{\gamma^2 + \gamma + 0.5} > 2\sqrt{\gamma^2} = 2\gamma$, which means that we need to choose the positive sign for $d_q > 0$ to be true. The value of $d_q$ that makes the derivative zero is therefore uniquely defined by:

$$d_q = d_e\,\gamma\frac{2\,\gamma + \sqrt{4\,\gamma^2 + 4\,\gamma + 2}}{2\,\gamma + 1} \geq d_e\gamma,$$

where the last inequality holds because $\sqrt{4\gamma^2 + 4\gamma + 2} = 2\sqrt{\gamma^2 + \gamma + 0.5} \geq 1$.

**2. Analyze the function at the boundaries of its' domain.**

To understand why this critical point corresponds to a maximum, we analyze the function $f(d_q)$ as $d_q$ at the boundaries of its domain.

**2a. $f(d_q)$ when $d_q = \gamma d_e$ $(d_q \geq \gamma d_e)$.**

$$\begin{aligned}
f(\gamma d_e) &= \frac{d_e \left(2(\gamma d_e)\gamma - 2d_e\gamma^2 + (\gamma d_e)\right)}{(\gamma d_e)(d_e + \gamma d_e)} \\
&= \frac{d_e \left(2d_e\gamma^2 - 2d_e\gamma^2 + \gamma d_e\right)}{(\gamma d_e)(d_e + \gamma d_e)} \\
&= \frac{d_e (\gamma d_e)}{(\gamma d_e)(d_e + \gamma d_e)} \\
&= \frac{d_e^2 \gamma}{(\gamma d_e)d_e(1 + \gamma)} \\
&= \frac{1}{1 + \gamma}.
\end{aligned}$$

**2b. $f(d_q)$ as $d_q \to \infty$ $(d_q \geq \gamma d_e)$.**

We want to evaluate the limit of $f(d_q)$ as $d_q$ approaches infinity:

$$\lim_{d_q \to \infty} f(d_q) = \lim_{d_q \to \infty} \frac{d_e(-2d_e\gamma^2 + (2\gamma + 1)d_q)}{d_e d_q + d_q^2}$$

Divide the numerator and the denominator by the highest power of $d_q$ in the denominator, which is $d_q^2$:

$$\lim_{d_q \to \infty} f(d_q) = \lim_{d_q \to \infty} \frac{d_e \left(-\frac{2d_e\gamma^2}{d_q^2} + \frac{2\gamma + 1}{d_q}\right)}{\frac{d_e}{d_q} + 1}$$

As $d_q \to \infty$, the terms $\frac{2d_e\gamma^2}{d_q^2}$, $\frac{2\gamma + 1}{d_q}$, and $\frac{d_e}{d_q}$ all approach 0. Thus,

$$\lim_{d_q \to \infty} f(d_q) = \frac{d_e(0 + 0)}{0 + 1} = 0$$

This means that as $d_q$ becomes very large, the function $f(d_q)$ approaches 0.

**2c. Showing that $f(d_q^*) \geq f(\gamma d_e)$.**

We want to show that $f(d_q^*) \geq f(\gamma d_e)$. Or equivalently, that $f(d_q^*) - f(\gamma d_e) \geq 0$. From Theorem 3 we know that $f(d_q^*) = 2\gamma \left(2\gamma + 1 - \sqrt{4\gamma^2 + 4\gamma + 2}\right) + 1$, and from 2a we know that $f(\gamma d_e) = \frac{1}{1+\gamma}$. After substitution and some algebraic manipulation, we get

$$\gamma \left(\frac{4\gamma^2 + 6\gamma + 3}{1 + \gamma} - 2\sqrt{4\gamma^2 + 4\gamma + 2}\right) \geq 0.$$

Since $\gamma > 0$, it suffices to show that

$$\frac{4\gamma^2 + 6\gamma + 3}{1 + \gamma} \geq 2\sqrt{4\gamma^2 + 4\gamma + 2}.$$

Squaring both sides of the above inequality and simplifying, we obtain the equivalent inequality

$$\left(\frac{4\gamma^2 + 6\gamma + 3}{1 + \gamma}\right)^2 \geq 4(4\gamma^2 + 4\gamma + 2).$$

After further algebraic manipulations (which we leave to the reader), we arrive at the inequality

$$(2\gamma + 1)^2 \geq 0.$$

Since $(2\gamma + 1)^2 \geq 0$ holds for all $\gamma$, and the previous steps are all equivalences, we conclude that

$$f(d_q^*) - f(\gamma d_e) \geq 0$$

for $0 < \gamma \leq 1$, and therefore,

$$f(d_q^*) \geq f(\gamma d_e).$$

**2d. Showing that $f(d_q^*) \geq \lim_{d_q \to \infty} f(d_q)$.**

We combine the results from 2a-2c to get

$$
\begin{aligned}
f(d_q^*) &\geq f(\gamma d_e) \\
&= \frac{1}{1+\gamma} \\
&\geq 0 \\
&= \lim_{d_q \to \infty} f(d_q).
\end{aligned}
$$

**2e. $f(d_q)$ as $d_q \to (\gamma d_e)^-$ $(d_q < \gamma d_e)$.**

Since we are approaching $\gamma d_e$ from the left, we have that $f(d_q) = d_q/(d_e + d_q)$. This function is continuous for $d_q < \gamma d_e$, so the limit is given by the direct substitution:

$$
\begin{aligned}
\lim_{d_q \to (\gamma d_e)^-} \frac{d_q}{d_e + d_q} &= \frac{\gamma d_e}{d_e + \gamma d_e} \\
&= \frac{\gamma d_e}{d_e(1+\gamma)} \\
&= \frac{\gamma}{1+\gamma}
\end{aligned}
$$

**2f. Showing that $f(\gamma d_e) \geq f(d_q)$ when $d_q < \gamma d_e$.** We start by noting that $f(\gamma d_e) = \frac{1}{1+\gamma} \geq \frac{\gamma}{1+\gamma} = \lim_{d_q \to (\gamma d_e)^-}$. Now it is sufficient to show that $f(d_q) = d_q/(d_q + d_e)$ is strictly decreasing for decreasing $d_q$, which we do by computing the derivative of $f(d_q)$ with respect to $d_q$ using the quotient rule:

$$
\begin{aligned}
f'(d_q) &= \frac{(d_q + \gamma)(1) - d_q(1)}{(d_q + \gamma)^2} \\
&= \frac{d_q + \gamma - d_q}{(d_q + \gamma)^2} \\
&= \frac{\gamma}{(d_q + \gamma)^2}.
\end{aligned}
$$

Since $\gamma > 0$ and $(d_q + \gamma)^2 > 0$ for all $d_q > 0$, we have $f'(d_q) > 0$ for all $d_q > 0$. This implies that the function $f(d_q)$ is strictly increasing on the interval $(0, \infty)$. Therefore, if $0 < c \leq b$, it must be the case that $f(c) \leq f(b)$. Moreover, since $c < b$, $f(c) < f(b)$. Thus, for any $b > 0$, $f(b) > f(c)$ for all $0 < c \leq b$. Now let $0 < d_q = c \leq \gamma d_e = b$.

**3. Combining everything (2a-2f)**

We have derived a unique critical point $d_q^* \geq \gamma d_e$ by setting the first derivative of $f(d_q)$ to zero. We have then shown that $f(d_q^*)$ is greater than or equal to $f(d_q)$ at the limits of its' domain when $d_q \geq \gamma d_e$. Finally, we show that $f(d_q^*) \geq f(\gamma d_e) \geq f(d_q)$ when $d_q < \gamma d_e$. Therefore, the value of $d_q$ that is the global maximum of $f(d_q)$ when $d_q > 0$ is:

$$\boxed{d_q^* = d_e\,\gamma\frac{2\,\gamma + \sqrt{4\,\gamma^2 + 4\,\gamma + 2}}{2\,\gamma + 1}}$$

$\square$

### A.3 Proof of Theorem 3

*Proof.* From Theorem 2 we have that

$$d_q^* = \frac{d_e \, \gamma \left(2\,\gamma + \sqrt{4\,\gamma^2 + 4\,\gamma + 2}\right)}{2\,\gamma + 1}$$

maximizes the function

$$f(d_q) = \frac{d_e \left(-2\,d_e\,\gamma^2 \; + \; (2\,\gamma + 1)\,d_q\right)}{d_q \,(d_e + d_q)}.$$

We wish to show that the maximum label accuracy given overlap, $f^*(\gamma) = f\left(d_q^*\right)$, is

$$2\,\gamma\!\left(2\,\gamma + 1 \; - \; \sqrt{4\,\gamma^2 + 4\,\gamma + 2}\right) \; + \; 1.$$

**1. Express $f(d_q)$ in terms of a dimensionless variable.**

Define

$$\delta \; = \; \frac{d_q}{d_e}.$$

Then

$$d_q \; = \; \delta\, d_e, \quad d_e + d_q \; = \; d_e\,(1 + \delta),$$

and

$$f(d_q) \; = \; f(\delta\, d_e) \; = \; \frac{d_e \left(-2\,d_e\,\gamma^2 + (2\,\gamma + 1)\,\delta\, d_e\right)}{(\delta\, d_e)\,(d_e + \delta\, d_e)} \; = \; \frac{-2\,\gamma^2 + (2\,\gamma + 1)\,\delta}{\delta\,(1 + \delta)}.$$

We can therefore write

$$f(\delta) \; = \; \frac{-2\,\gamma^2 \; + \; (2\,\gamma + 1)\,\delta}{\delta\,(1 + \delta)}.$$

**2. Identify the optimal dimensionless query length $\delta^*$.**

From Theorem 2, we know that

$$d_q^* \; = \; \frac{d_e\,\gamma\left(2\,\gamma \; + \; \sqrt{4\,\gamma^2 + 4\,\gamma + 2}\right)}{2\,\gamma + 1}.$$

Dividing both sides by $d_e$ gives

$$\delta^* \; = \; \frac{d_q^*}{d_e} \; = \; \gamma\,\frac{2\,\gamma \; + \; \sqrt{4\,\gamma^2 + 4\,\gamma + 2}}{2\,\gamma + 1}.$$

We need to show that

$$f\left(\delta^*\right) \; = \; 2\,\gamma\!\left(2\,\gamma + 1 - \sqrt{4\,\gamma^2 + 4\,\gamma + 2}\right) \; + \; 1.$$

**3. Compute $f(\delta^*)$ explicitly.**

Let

$$N(\delta) \; = \; -2\,\gamma^2 \; + \; (2\,\gamma + 1)\,\delta, \quad D(\delta) \; = \; \delta\,(1 + \delta).$$

Then $f(\delta) = \frac{N(\delta)}{D(\delta)}$.

1. *Numerator at $\delta^*$.*

$$N(\delta^*) = -2\gamma^2 + (2\gamma+1)\delta^* = -2\gamma^2 + (2\gamma+1)\Big[\gamma \frac{2\gamma + \sqrt{4\gamma^2+4\gamma+2}}{2\gamma+1}\Big].$$

Inside the brackets, $(2\gamma+1)$ cancels:

$$N(\delta^*) = -2\gamma^2 + \gamma\left(2\gamma+\sqrt{4\gamma^2+4\gamma+2}\right) = -2\gamma^2 + 2\gamma^2 + \gamma\sqrt{4\gamma^2+4\gamma+2} = \gamma\sqrt{4\gamma^2+4\gamma+2}.$$

2. *Denominator at $\delta^*$.*

$$D(\delta) = \delta(1+\delta).$$

Hence,

$$D(\delta^*) = \delta^*\left(1+\delta^*\right) = \Big[\gamma\frac{2\gamma+\sqrt{4\gamma^2+4\gamma+2}}{2\gamma+1}\Big]\Big[1 + \gamma\frac{2\gamma+\sqrt{4\gamma^2+4\gamma+2}}{2\gamma+1}\Big].$$

The second bracket becomes a single fraction:

$$1 + \gamma\frac{2\gamma+\sqrt{4\gamma^2+4\gamma+2}}{2\gamma+1} = \frac{(2\gamma+1) + \gamma\left(2\gamma+\sqrt{4\gamma^2+4\gamma+2}\right)}{2\gamma+1}.$$

Combining, we get

$$D(\delta^*) = \gamma\frac{2\gamma+\sqrt{4\gamma^2+4\gamma+2}}{2\gamma+1} \times \frac{(2\gamma+1) + 2\gamma^2 + \gamma\sqrt{4\gamma^2+4\gamma+2}}{2\gamma+1}.$$

So

$$D(\delta^*) = \gamma\frac{\left(2\gamma+\sqrt{4\gamma^2+4\gamma+2}\right)\left(2\gamma+1+2\gamma^2+\gamma\sqrt{4\gamma^2+4\gamma+2}\right)}{(2\gamma+1)^2}.$$

3. *Form the ratio.* Thus,

$$f(\delta^*) = \frac{N(\delta^*)}{D(\delta^*)} = \frac{\gamma\sqrt{4\gamma^2+4\gamma+2}}{\gamma\frac{\left(2\gamma+\sqrt{4\gamma^2+4\gamma+2}\right)\left(2\gamma+1+2\gamma^2+\gamma\sqrt{4\gamma^2+4\gamma+2}\right)}{(2\gamma+1)^2}}.$$

Cancel the common factor $\gamma$, invert the denominator and multiply:

$$f(\delta^*) = \frac{\sqrt{4\gamma^2+4\gamma+2}\,(2\gamma+1)^2}{\left(2\gamma+\sqrt{4\gamma^2+4\gamma+2}\right)\left(2\gamma+1+2\gamma^2+\gamma\sqrt{4\gamma^2+4\gamma+2}\right)}.$$

You can verify by direct expansion (or by a symbolic algebra tool which we provide in the supplementary material) that

$$\frac{\sqrt{4\gamma^2+4\gamma+2}\,(2\gamma+1)^2}{\left(2\gamma+\sqrt{4\gamma^2+4\gamma+2}\right)\left(2\gamma+1+2\gamma^2+\gamma\sqrt{4\gamma^2+4\gamma+2}\right)} = 2\gamma\left(2\gamma+1-\sqrt{4\gamma^2+4\gamma+2}\right) + 1.$$

Thus

$$f(\delta^*) = 2\gamma\left(2\gamma+1-\sqrt{4\gamma^2+4\gamma+2}\right) + 1,$$

which proves that

$$f^*(\gamma) = f(d_q^*) = 2\gamma\left(2\gamma+1-\sqrt{4\gamma^2+4\gamma+2}\right) + 1.$$

Hence, Eq. 11 holds, completing the proof. $\qquad\square$

## A.4 Empirical Analysis of Uniform Relative Offset Distribution Between Events and Overlapping Segments

We empirically verify that the relative offset between events and overlapping query segments can be modeled well by a uniform distribution. An event is denoted by $e = (a_e, b_e, c_e)$, where $a_e$ is the start time, $b_e$ is the end time, and $c_e$ is the class, where $c_e \in \{\text{"dogs"}, \text{"baby"}\}$. Similarly, a query segment is denoted by $q = (a_q, b_q)$. We define the event distribution using the labeled start times of different sound event classes from the NIGENS Trowitzsch et al. (2019) dataset, but we fix the event length $d_e$ to the median event length of the respective sound class to respect the deterministic event length assumption. We then verify that the relative offset between the events and the overlapping segments, defined as $b_q - a_e$, is uniform over the range $[0, d_e + d_q]$. To simulate realistic scenarios that maintain a reasonable label accuracy, we let $d_q \in \{\frac{d_e}{10}, d_e, 10d_e\}$. That is, the query segment is not larger than 10 times the median event length. Note that if $d_q \gg d_e$ then the uniform relative offset assumption is not expected to hold, but that also means that the label accuracy will be very low which is not wanted in practice. We present the results in Figure 19. For both sound event classes the distribution looks flat for all three choices of $d_q$, meaning that it can be modeled well by a uniform distribution, verifying that this is a plausible assumption in practical scenarios.

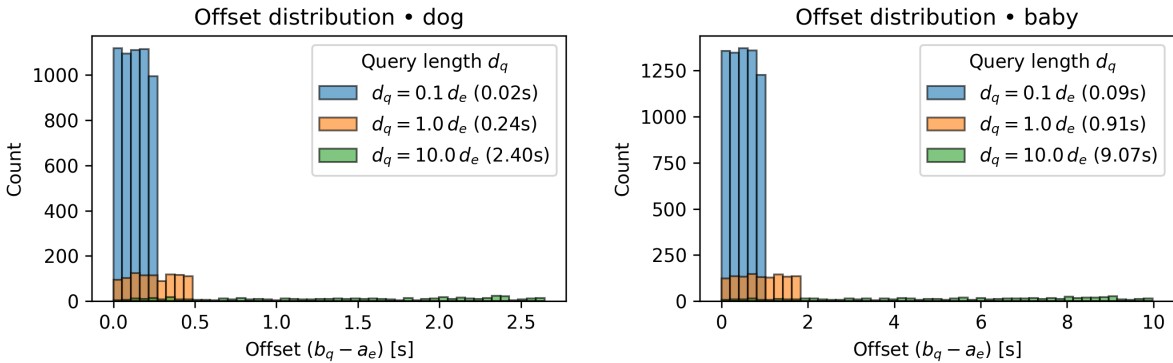

Figure 19: The distribution of relative offsets between events and overlapping query segments for dog (left) and baby (right) sound event classes from the NIGENS Trowitzsch et al. (2019) dataset. We use the annotated relative start times of the events, and fix the event length $d_e$ to the median event length for the respective event class. The distribution looks flat and can be modeled well with a uniform distribution.

## A.5 Empirical Analysis of Theory

In an attempt to empirically validate the theory we have compared the weakly labeled version of AudioSet Hershey et al. (2017) with the strongly labeled version Hershey et al. (2021). The weakly labeled version of AudioSet uses 10 second segments, corresponding to $d_q = 10$. A subset of AudioSet has been strongly labeled by indicating start and end times of the weakly labeled events. For each strongly labeled event, we compare the strong label to the weak label to compute the accuracy. That is, if the weakly labeled segment of length $d_q$ indicates the presence of an event, and the corresponding strong label for that event has length $d_e$, then the accuracy is $\frac{d_e}{d_q}$ for that event. We compute this accuracy for all sound events corresponding to the "Animal" class, and take the average. The theoretical accuracy for a given annotator criterion $\gamma$ is derived by taking the numerical average over the event lengths for the "Animal" class to estimate the numerical integration over the event length distribution presented Eq. (13).

The results are shown in Figure 20. Note that the query segment is not chosen to maximize label accuracy as in previous analysis. Since $d_q = 10$ it is longer than most 'Animal' events in AudioSet and restricts the maximum event length that can be annotated to $d_e \leq d_q$ as can be seen in the right panel of Figure 20. In the left panel of Figure 20 we see that the label accuracy of the weakly labeled version of AudioSet falls within the range predicted by the theory for $\gamma \in (0, 1]$. Assuming that the theory is correct would indicate that $\gamma = 0.26$ is the presence criterion that best models the weak labeling process of AudioSet. While these

results do not reject the proposed theory we would need to empirical estimate $\gamma$ based on real annotators to properly validate it. This is considered as out of scope for this paper, but would be interesting future work.

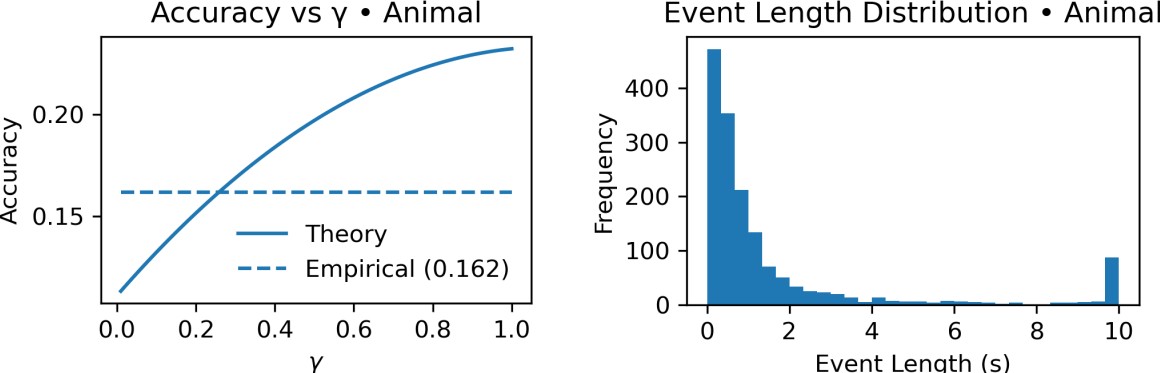

Figure 20: The theoretical prediction of the label accuracy for different $\gamma$ (left) when averaging over the event length distribution (right) for the animal events in the strongly labeled subset of AudioSet. The empirical accuracy (dashed blue line) indicates the label accuracy that was derived by comparing the weakly labeled version of AudioSet with the strongly labeled version. The empirical label accuracy falls within the range predicted by the theory.

