# OpenReview forum: "The Accuracy Cost of Weakness: A Theoretical Analysis of Fixed-Segment Weak Labeling for Events in Time"
_TMLR — Accepted by TMLR_

### Review · Reviewer_ZKzU · 2025-03-25

**Summary Of Contributions:**

**Summary**

This paper presents a theoretical analysis of fixed-length (FIX) weak labeling for event detection in time series data, particularly focusing on audio recordings. The authors model the accuracy and annotation cost of this common labeling approach where annotators assign presence/absence labels to fixed-length segments. They compare this to an oracle (ORC) method that uses true event boundaries and derive closed-form expressions for label accuracy and annotation cost for both methods. The theoretical framework is validated through simulations, showing that in most realistic scenarios, the oracle method outperforms the fixed-length approach in both accuracy and cost. The paper provides a foundation for developing adaptive weak labeling strategies that mimic the oracle process.

**Audience:**

Yes

**Broader Impact Concerns:**

The authors have provided discussions about several practical implications.

**Claims And Evidence:**

Yes

**Requested Changes:**

Please refer to the weakness part for details. In brief, it would be better to provide more comparisons or at least discussions for the application in real-world annotation data or other multimodal labeling noise fields.

**Strengths And Weaknesses:**

**Strengths**

- The paper presents a novel theoretical framework for analyzing weak labeling methods in temporal event detection.
- Extensive simulation studies effectively validate the theoretical results across various scenarios, including different annotator criteria, event length distributions, and multiple events.
- The paper is well-structured and clearly written, with helpful visualizations and mathematical derivations.

**Weaknesses**

- While the paper extensively validates the theoretical results through simulations, it lacks real-world experimental validation. The assumptions made in the theoretical model (e.g., uniformly distributed events, deterministic annotator behavior) may not fully capture the complexities of real annotation tasks. Real annotators may exhibit variability in their perception, attention, and judgment that isn't captured by the current model. If possible, testing the framework on actual human annotation data would strengthen the paper's claims.
- The paper compares FIX and ORC methods, but could benefit from comparisons with other weak labeling strategies, such as active learning approaches or semi-automated methods. Such comparisons would provide a more comprehensive understanding of the relative advantages of different labeling strategies in the accuracy-cost space.
- The theoretical analysis assumes that events are spaced at least
 apart for multiple events, which may not hold in high-density scenarios. While the authors do perform simulations that relax this assumption, the theoretical implications of violating this assumption are not fully explored in the mathematical framework.
- Some aspects of the analysis could be more explicit about their generalizability. For instance, while the framework is developed with audio events in mind, more discussion on how the results extend to other domains with temporal events (e.g., video analysis, physiological signals, financial time series) would enhance the paper's impact. I am wondering could the proposed method be applied to multimodal temporal classification tasks, like audio-visual event localization (AVEL) or video parsing (AVVP). Some works in AVVP also aim to address the label noise or noise rate, such as (Joint-Modal Label Denoising for Weakly-Supervised Audio-Visual Video Parsing; Advancing Weakly-Supervised Audio-Visual Video Parsing via Segment-Wise Pseudo Labeling). If the proposed method is not applicable to these tasks, it would also be better to add these related works in the introduction of label noise in Sec. 1.

---

> ### Author Response · Authors · 2025-07-04
> **Strengthening Assumptions and Adding Empirical Validation: Response to Your Review**
>
> Dear Reviewer,
>
> Thank you for your constructive feedback. We appreciate you noting the novel theoretical framework, simulation effectiveness, and clear structure. We have addressed your suggestions, which have significantly strengthened the paper.
>
> To improve clarity and integrate these changes, we have rewritten the Discussion (Section 8) for a more logical flow. Below, we detail how we addressed each point.
>
> **On validation and assumptions:** This was a primary concern we share. To address it, we made two major additions:
> 1.  We relaxed the global uniform event distribution assumption to a more realistic "local" one. We validate this on the NIGENS dataset in the new Appendix A.4.
> 2.  For real-world validation of the theory, we added Appendix A.5, which compares our model's predictions to measured accuracy on AudioSet. This is referenced in the rewritten Discussion.
>
> **On weak labeling strategies:** Our revised Discussion now frames our work as a theoretical foundation for more advanced adaptive methods (e.g., active learning). The "Implications for Practical Annotation" subsection clarifies how our analysis of the FIX-ORC gap defines the performance spectrum for these methods.
>
> **On spaced-apart events:** You correctly note this is a simplification. Our simulations (Section 5) relax this by merging overlapping events, showing our theory serves as a robust upper bound even in dense scenarios. We note this for future theoretical work.
>
> **On generalizability and multimodal applications:** We added a "Generalization and Future Directions" subsection to the Discussion, which discusses applicability to other temporal domains (e.g., physiological signals) and its extension to higher dimensions (e.g., images). We also incorporated your references into Related Work (Section 7.1), distinguishing our segment label noise from modality-specific noise.
>
> We believe these revisions substantially improve the paper. A summary of changes is below. Thank you again for your feedback.
>
> ---
> ### **Summary of Changes**
> *(in response to feedback from all reviewers)*
>
> **Discussion Section Restructured (Section 8, pp. 19-20)**
> - **Change:** Rewrote the Discussion with four clear subheadings.
> - **Motivation:** To improve logical flow and integrate new analyses addressing reviewer points (Reviewers ZKzU, cKD4).
>
> **Appendices Added for Empirical Validation (Appendix A.4 & A.5, pp. 33-34)**
> - **Change:** Added two new appendices with empirical analysis.
> - **Motivation:** To provide real-world validation and test core assumptions (Reviewer ZKzU). A.4 validates the local uniform assumption; A.5 validates the main theory on AudioSet.
>
> **Data Distribution Assumption Refined (Sections 2.1 & 4.2, pp. 3 & 7)**
> - **Change:** Relaxed the global uniform event assumption to a more realistic local one.
> - **Motivation:** To strengthen the theoretical foundation with a more defensible assumption (Reviewer ZKzU).
>
> **Related Work Contextualized (Section 7.1, p. 17)**
> - **Change:** Expanded noisy label discussion to include multimodal references and distinguish noise types.
> - **Motivation:** To position our work in a broader context as suggested by Reviewer ZKzU.
>
> **Annotator Model Rationale Clarified (Section 2.2, p. 4)**
> - **Change:** Added a sentence clarifying our annotator model rationale.
> - **Motivation:** To clarify modeling choices per Reviewer cKD4's question.
>
> Sincerely,
> The Authors

---

### Review · Reviewer_cKD4 · 2025-06-06

**Summary Of Contributions:**

This paper theoretically analyzes the accuracy and cost trade-offs of fixed-segment weak labeling (FIX) for temporal event annotation, comparing it with an oracle method (ORC) that uses true event boundaries. The authors model how segment length affects label accuracy and annotation effort, defining key concepts like event fraction overlap $h(e, q)$ and a presence criterion $\gamma$ to quantify annotator sensitivity. Through closed-form expressions and simulations, they show that FIX introduces segment label noise, with shorter segments improving accuracy at higher costs, while longer segments reduce costs but degrade accuracy.

**Audience:**

Yes

**Broader Impact Concerns:**

Not Applicable.

**Claims And Evidence:**

Yes

**Requested Changes:**

Please refer to the weaknesses above.

**Strengths And Weaknesses:**

- **Strengths**
    1. This work derives expected label accuracy for FIX and ORC under single/deterministic/multiple stochastic event scenarios, the analysis of which is very meticulous.
    2. It proves that ORC achieves perfect accuracy with minimal segments, serving as an upper bound for FIX.
    3. It provides a theoretical foundation for adaptive weak labeling strategies mimicking ORC, such as change-point detection and active learning.
    5. The paper is well-structured, with clear logic from problem statement to experimental verification.

- **Weaknesses**
    - My main concern is that this work merely analyzes the mathematical properties of the accuracy distribution under the scenario of temporal event annotation, and the conclusions are simple and intuitive. However, it does not highlight how to guide algorithm design based on the conclusions, offering limited guidance for machine learning algorithms and experimental design. Perhaps the paper needs to propose a specific active learning algorithm to validate the proposed modeling, rather than merely conducting simulation experiments.
    - It may be unreasonable to determine presence or absence by the proportion of the overlapping part with $d_e$. Why not model it with $|e\cap q| / q$ ?Some discussions can be added.
    - Figure 4 could be polished up. It is recommended to make the top panel dynamic (i.e., showing two scenarios of q and e overlapping) and use dashed lines to correspond with the bottom panel.

---

> ### Author Response · Authors · 2025-07-04
> **Clarifying Scope and Practical Implications: Response to Your Review**
>
> Dear Reviewer,
>
> Thank you for your thoughtful review. We are grateful that you recognized the meticulousness of our analysis and its value as a theoretical foundation for adaptive methods. Your feedback has helped us clarify the practical contributions and scope of our work.
>
> To accommodate these clarifications, we have substantially restructured the **Discussion (Section 8)**, and we believe the revised manuscript is much stronger as a result. Below, we address the weaknesses you identified.
>
> **On Algorithmic Guidance and Scope:** We agree that theory should guide practice. Our paper achieves this in two ways:
>
> 1.  **Direct Guidance for FIX Labeling:** Our work provides direct, actionable guidance for the common FIX labeling method. **Theorems 2 and 4** offer a clear rule of thumb for selecting the optimal segment length and estimating annotation cost, informing practical protocol design.
>
> 2.  **A Theoretical Foundation for Adaptive Methods:** While we don't propose a new adaptive algorithm, our primary contribution is establishing the theoretical landscape that motivates them. By quantifying the performance gap between FIX and the idealized ORC method, we define the "accuracy cost of weakness" that adaptive algorithms aim to minimize. Our restructured **Discussion (Section 8)** now includes "Implications for Practical Annotation" (p. 19), framing our work as the theoretical motivation for these advanced strategies.
>
> **On Modeling the Presence Criterion (`γ`):** To clarify, the presence criterion, `h(e,q) ≥ γ`, is not a rule stating when an annotator should indicate presence of an event, but rather a *model* of their perceptual ability to recognize an event. We see two primary recognition scenarios:
>
> 1.  **Relative Duration (Non-stationary events):** For structured sounds like a birdsong, recognition requires perceiving a sufficient portion of its unique structure. This is inherently relative to the event's duration (`d_e`). Our model, `|e∩q| / d_e`, is designed for this common case.
> 2.  **Absolute Duration (Stationary events):** For sounds like an engine hum, an annotator might only need to hear a fixed amount of time (e.g., 0.5 seconds).
>
> Our choice is motivated by the prevalence of non-stationary events in challenging annotation tasks. We have added a sentence to **Section 2.2** (p. 4) to state this rationale explicitly.
>
> **On Polishing Figure 4:** **Figure 4** is a precise visual aid for the mathematical derivation of the integral in **Equation (6)**. It illustrates how the accuracy function evolves as an event sweeps across a fixed query segment. We believe its current form is a direct and accurate depiction of this mathematical concept.
>
> We are confident these revisions have substantially improved the manuscript. A summary of all changes is below. Thank you again for your valuable feedback.
>
> ---
> ### **Summary of Changes**
> *(in response to feedback from all reviewers)*
>
> **Discussion Section Restructured (Section 8, pp. 18-20)**
> - **Change:** Rewrote the Discussion with four clear subheadings.
> - **Motivation:** To improve logical flow and integrate new analyses addressing reviewer points (Reviewers ZKzU, cKD4).
>
> **Appendices Added for Empirical Validation (Appendix A.4 & A.5, pp. 33-34)**
> - **Change:** Added two new appendices with empirical analysis.
> - **Motivation:** To provide real-world validation and test core assumptions (Reviewer ZKzU). A.4 validates the local uniform assumption; A.5 validates the main theory on AudioSet.
>
> **Data Distribution Assumption Refined (Sections 2.1 & 4.2, pp. 3 & 7)**
> - **Change:** Relaxed the global uniform event assumption to a more realistic local one.
> - **Motivation:** To strengthen the theoretical foundation with a more defensible assumption (Reviewer ZKzU).
>
> **Related Work Contextualized (Section 7.1, p. 17)**
> - **Change:** Expanded noisy label discussion to include multimodal references and distinguish noise types.
> - **Motivation:** To position our work in a broader context as suggested by Reviewer ZKzU.
>
> **Annotator Model Rationale Clarified (Section 2.2, p. 4)**
> - **Change:** Added a sentence clarifying our annotator model rationale.
> - **Motivation:** To clarify modeling choices per Reviewer cKD4's question.
>
> Sincerely,
> The Authors

---

### Review · Reviewer_Gdrx · 2025-08-01

**Summary Of Contributions:**

The paper considers the problem of human annotation and possible introduced noise in the annotation due to the process itself. Authors consider specifically problem of segmentation annotation (annotate that particular segment of signal is classified as particular class: e.g. audio/video event segmentation) where one of the approaches to human annotation is using fixed segments and giving these fixed segments for human labeling (assign if particular event is present in that segment) only to reduce annotation costs. While this annotation process reduces the cost of human labeling, authors discuss that this process introduces segment noise: missing some events, wrongly annotating some events, etc. The main question authors consider is what will be the expected segment noise error in the annotation and how to estimate the best segment size to use in the annotation process if we know e.g. some statistic on the average event duration.

The main contribution is giving math derivations on the expected segment noise and finding the optimal segment size (minimizing this expected error) under some assumptions (the best e.g. if we know the average event duration or min / max duration we can detect the event). Furthermore authors are doing data simulation showing that their derivations under some assumptions are pretty general and well describe more complicated scenarios including some real data, e.g. AudioSet.

Authors give a nice discussion at the end of the paper on how to use, interpret their results and kind of pushing to revisit our human annotation process in future with adaptive and active learning techniques e.g.

**Audience:**

Yes

**Broader Impact Concerns:**

No any concerns based on the content and derivations done in the paper.

**Claims And Evidence:**

Yes

**Requested Changes:**

**Some conceptual questions:**
- I found it strange that we are still doing fixed segment annotations. Already 10 years ago Google presented their tool (and as far as I know a lot of places are doing the same now) where they use a model to annotate image segmentation and then give this to human annotators to correct boundaries and correct class assignment. This speeds up annotation and of course makes it cheaper as it is now just a correction. The same e.g. done also for speech recognition. I am not familiar with the sound segmentation, but I am really surprised that this is not done yet there / or not widespread. Could you comment on this? Because otherwise I don't understand the point of finding the best segment size for the task if the process is more adaptive and based on automatic segmentation in the first place.
- It is great that you can derive the exact estimation of the segment noise, but at the same time I feel it is kind of obvious that this scheme introduces large noise / error into annotation and yep, optimal segment will depend on the de/dq and in general on the event duration / perception for particular task. Being obvious and simple doesn't mean to me it is not valuable and we may need to do it and show due to poor practices. But is it really that community (as I am a bit different domain where we don't do such segmentation for human annotation) is actively doing it due to negligence that there will be high error?

**Some questions:**
- what happens if two different events occurred at the same time? Do you consider only one class assignment? or is it allowed to assign two classes for human annotators? How will this impact your derivations?
- Assumption "events are uniformly distributed locally in time" seems too restrictive to me and this is not what happens in real data. How does this affect the derivations and generalization of the exact formulas you got?
- "This is arguably the simplest data distribution to annotate, and the results can therefore be viewed as an upper bound on the expected label accuracy for any more complex data distribution." - I quite don't get why this will be upper bound? Is this in the sense of worst case scenario? I would argue the opposite.
- Figure 4: are you sure that the central place of segment A will be at 0.5? I think this is wrong and the picture thus is misleading (though in appendix derivations are good to me, e.g. in Table 2).
- Could you confirm that your empirical simulations well describe math prediction for the multiclass data where several events can happen in the long audio? I think I miss this more complicated (what we have in practice) scenario a bit.

**Some suggestions:**
- While reading I was most of the time confused between weak labels as annotations and weak labels as in pseudo-labeling (and automatic annotation). All discussions with related works made me confused which piece of the weak labels you are focusing on.
- Maybe it is good to move some discussion sections in the beginning and give the main outcomes to people first and then go into math and empirical simulations. I feel that the discussion section contains a lot of important pieces as the outcome of your work which most people need to know when they read the paper w/o going into derivations. It feels that the main message is kind of lost in the paper as it is hidden at the very end.

**Strengths And Weaknesses:**

**Strengths**

- Math derivations on the optimal segment size to use for human annotations to reduce segment noise
- Empirical simulations to show that math even under assumptions well describes more realistic data, including some simulations with real data like AudioSet.
- Discussion of consequences of the estimated noise and what the best practices we should work on for human annotation in the segmentation domain.

**Weaknesses**

(see requested changes for more details)

- I have conceptual issues with the method authors are trying to analyse and point out that it is not a good way to do human annotation. I think we have adopted a long time ago segmentation process using ML model first and then fixing the segmentation with both labeling at the same time.
- (maybe I just missed it) Empirical justification for the real data where we have long audio with many events occurring in it: showing that derivations will be well aligned with such data distribution.


**Note** I did a pass over all proofs (some not exactly deep on the derivations exactly but all transitions I have double checked) and all seems good to me w/o obvious errors. Also the paper itself is well written - did not spot any typos so far.

---

> ### Author Response · Authors · 2025-08-19
> **Clarifying Motivation and Addressing Conceptual Questions**
>
> Dear Reviewer,
>
> Thank you for your thorough and constructive review. We appreciate your positive assessment of our derivations and simulations. Your insightful conceptual questions have helped us clarify the practical motivation for our work, and we detail our planned revisions below.
>
> **On the relevance of analyzing FIX labeling:**
>
> You raise a crucial point regarding the practicality of fixed-segment (FIX) annotation. While we agree that adaptive methods are superior, FIX is not an outdated technique; it remains a de facto standard for large-scale annotation due to its simplicity and scalability, as seen in foundational datasets like AudioSet, CHIME, MAESTRO Real, and OpenMIC-2018.
>
> Our work provides the first formal analysis of this widespread practice. You correctly note that FIX introduces "obvious" noise; our contribution is to move beyond this intuition by quantifying this noise mathematically. This provides the necessary theoretical motivation and baseline needed to justify developing the more advanced, adaptive methods you suggest.
>
> To clarify this, we will revise the Introduction (Sec. 1) to emphasize the prevalence of FIX by citing these key datasets earlier. We will also revise the Discussion (Sec. 8) to frame our work as the theoretical baseline that motivates adaptive methods.
>
> **On your specific questions:**
>
> 1. **Overlapping events of different classes:** Our analysis is framed within a multi-pass binary labeling setting, where an annotator focuses on the presence or absence of a single event class at a time. We clarify this in Section 2 (p. 3), noting that overlapping events of the same class are treated as a single, longer event.
>
> 2. **Uniform distribution assumption:** This is an excellent point, also raised by other reviewers. In response, we have already relaxed the global uniform assumption to a more defensible "local" uniform assumption. We have also added Appendix A.4 (p. 33), which provides an empirical validation of this assumption using the NIGENS dataset, showing it holds well in practical scenarios. We will ensure our response clearly points to these revisions in Sections 2.1 & 4.2 and the new appendix.
>
> 3. **Upper bound on accuracy:** We apologize for the lack of clarity. Our reasoning is that a recording with a single, deterministic-length event is arguably the simplest possible annotation scenario. Any additional complexity—such as multiple events, event-length variability, or event density—introduces more opportunities for error. Therefore, the accuracy achieved in this idealized case should be higher than in more complex, realistic distributions, thus serving as a theoretical upper bound. We will add a sentence to Section 4.2 (p. 6) to make this logic explicit.
>
> 4. **Figure 4:** Thank you for this observation. The figure is intended to be a precise visualization of the function F(et, q, γ) being integrated in Equation 7, plotted against the event's end time t. It is not meant to suggest a general symmetry or that the peak accuracy is always 0.5. To prevent misinterpretation, we will revise the caption of Figure 4 (p. 7) to clarify that it illustrates the evolution of the accuracy function as the event sweeps across the query segment.
>
> 5. **Simulations for complex scenarios:** We did in fact run simulations for these more complex cases. We will explicitly direct you to Section 6.1.3 and Figures 11 and 12 (p. 14-15), which show results for recordings with 30 and 50 events, respectively. These results demonstrate that our theory serves as a robust upper bound even when the event density is high and the assumption of spaced-apart events is violated.
>
> **On your suggestions:**
>
> 1. **Clarifying "weak labels":** This is a valuable suggestion. To avoid confusion between weak labels from annotation versus pseudo-labels from models, we will add a sentence in the introduction to explicitly define the scope of "weak labeling" for our paper, focusing on the temporal ambiguity from the annotation process (segment label noise).
>
> 2. **Moving the discussion:** We agree that highlighting the main outcomes earlier is beneficial. While a full restructuring is difficult, we will add a concise paragraph at the end of the introduction summarizing the key practical implications of our findings. This will ensure readers grasp the main take-home messages without having to wait until the end.
>
> Thank you again for your constructive feedback, which has significantly improved our manuscript.
>
> Sincerely,
> The Authors

---

> > ### Comment · Reviewer_Gdrx · 2025-08-22
> > **Reply to clarifications**
> >
> > Dear Authors,
> >
> > Thanks for clarifications. Please find my followup thoughts and suggestions below:
> >
> > > You raise a crucial point regarding the practicality of fixed-segment (FIX) annotation. While we agree that adaptive methods are superior, FIX is not an outdated technique; it remains a de facto standard for large-scale annotation due to its simplicity and scalability, as seen in foundational datasets like AudioSet, CHIME, MAESTRO Real, and OpenMIC-2018.
> >
> > Well, if this is still de-facto standard, I think you really need to revisit introduction a bit to stress it more and clarify. References to adaptive labeling where it is applied sparingly and to still de-facto standard for FIX will make introduction and motivation strong. For references you pointed (AudioSet, CHIME, MAESTRO Real, and OpenMIC-2018.) - they are a bit old, except MAESTRO Real - maybe good idea to check recent NeurIPS datasets and benchmarks papers and some data companies to see if this is still the case of using FIX to make your argument stronger + inspire future papers and research on switching to more adaptive processes or develop / benchmark them rigorously.
> >
> > > Our work provides the first formal analysis of this widespread practice. You correctly note that FIX introduces "obvious" noise; our contribution is to move beyond this intuition by quantifying this noise mathematically. This provides the necessary theoretical motivation and baseline needed to justify developing the more advanced, adaptive methods you suggest.
> >
> > Yep, agree.
> >
> > > Overlapping events of different classes: Our analysis is framed within a multi-pass binary labeling setting, where an annotator focuses on the presence or absence of a single event class at a time. We clarify this in Section 2 (p. 3), noting that overlapping events of the same class are treated as a single, longer event.
> >
> > But what happens if there are events of different classes which happens at the same time? On which one annotator should focus? Do you have two different annotators which focus on two different labels - if so, then how you operate that you have two events of different labels which overlap in time?
> >
> > > Simulations for complex scenarios: We did in fact run simulations for these more complex cases. We will explicitly direct you to Section 6.1.3 and Figures 11 and 12 (p. 14-15), which show results for recordings with 30 and 50 events, respectively. These results demonstrate that our theory serves as a robust upper bound even when the event density is high and the assumption of spaced-apart events is violated.
> >
> > Thanks for pointing out. Sorry that I missed it before.
> >
> > > Clarifying "weak labels": This is a valuable suggestion. To avoid confusion between weak labels from annotation versus pseudo-labels from models, we will add a sentence in the introduction to explicitly define the scope of "weak labeling" for our paper, focusing on the temporal ambiguity from the annotation process (segment label noise).
> >
> > > Moving the discussion: We agree that highlighting the main outcomes earlier is beneficial. While a full restructuring is difficult, we will add a concise paragraph at the end of the introduction summarizing the key practical implications of our findings. This will ensure readers grasp the main take-home messages without having to wait until the end.
> >
> > Thanks! This all makes sense to me!
> >
> > Looking forward for the updated manuscript.
> >
> > Reviewer.

---

> > > ### Author Response · Authors · 2025-08-29
> > > **Response by authors**
> > >
> > > Dear Reviewer,
> > >
> > > Thank you again for your thoughtful follow-up. We very much appreciate your suggestions for strengthening the motivation and references, and we will integrate them into the final manuscript revision.
> > >
> > > In addition to the points raised earlier, we would like to emphasize one more motivation for why FIX segmentation is still widely used: for many sound sources it is inherently difficult (or even impossible) to define exact start and end times. A car passing by, or background noise fading in/out, does not have sharp temporal boundaries. Moreover, audio scenes often contain overlapping sources, making it hard to pre-segment into coherent events. These challenges make FIX a practical default even when adaptive methods are conceptually appealing.
> > >
> > > Regarding your question about overlapping events of different classes: in practice, annotation is often handled in a multi-pass setup where each class is annotated independently (and potentially by different annotators). Thus, two overlapping events of different classes are not mutually exclusive but can both be assigned within the same temporal window. We will clarify this explicitly in Section 2.
> > > We will also take care over the next week to incorporate your suggestions, especially reinforcing the motivation with more recent dataset references and adding a sharper framing of FIX vs. adaptive annotation, into the revised manuscript.
> > >
> > > Thank you again for helping us improve the clarity and impact of our work.
> > >
> > > Sincerely,
> > > The Authors

---

> > > > ### Comment · Reviewer_Gdrx · 2025-09-08
> > > > **Question about revision**
> > > >
> > > > Dear Author,
> > > >
> > > > Any update on the revision? Right now I don't see any updates from July version.
> > > >
> > > > Thanks,
> > > >
> > > > Reviewer.

---

> > > > > ### Author Response · Authors · 2025-09-09
> > > > > **PDF is now updated**
> > > > >
> > > > > Dear reviewer, we have now uploaded a revised version of the paper. We have highlighted major changes in the text, and added a description of the revision in the system.
> > > > >
> > > > > Thank you!

---

### Decision · Action_Editor_J3Vf · 2025-09-18

**Recommendation:** Accept with minor revision

**Additional Comments:**

Please explain why people not using advance annotation process yet widely, in the final version.

**Audience:**

Yes

**Audience Explanation:**

This paper belongs to the machine learning area, hence TMLR's audience will be interested.

**Claims And Evidence:**

Yes

**Claims Explanation:**

The claims of this paper are supported by a combination of theoretical derivations, simulations, and partial empirical validation. This paper makes a compelling case for the limitations of fixed-segment weak labeling and the potential benefits of adaptive methods.

---

> ### Author Response · Authors · 2025-09-19
> **Question about final version upload and author order**
>
> This is wonderful news! Thank you both to reviewers and editors for your work.
>
> We have now uploaded a camera ready version. A couple of practical comments:
>
> * In the last revision, uploaded September 9, we added the following section to the introduction (among a few other changes in response to reviewer Gdrx). We think that this is an explanation on why people are not using advanced annotation processes yet widely. We hope that you agree.
>
> "While adaptive annotation methods have been explored in some domains, fixed-segment (FIX) annotation
> remains the de facto standard for large-scale sound event datasets due to its simplicity and scalability. Foun-
> dational datasets such as AudioSet (Gemmeke et al., 2017a), CHIME (Foster et al., 2015b), OpenMIC-2018
> (Humphrey et al., 2018a), YBSS-200 (Singh & Joshi, 2019), and SONYC (Bello et al., 2019), as well as more
> recent datasets such as VGG Sound (Chen et al., 2020), MATS (Morato & Mesaros, 2021), and MAESTRO
> Real (Morato et al., 2023) continue to rely on fixed segments. A practical motivation is that for many sound
> sources, precise boundaries are inherently ambiguous: a passing car or fading background noise lacks sharp
> onsets and offsets, and audio scenes frequently contain overlapping sources. These challenges make FIX
> annotation a practical default even when adaptive approaches are conceptually appealing."
>
> * We have discussed the author order and Would like to change author number 2 to be the last author. We have made this change in the submitted camera ready version, but the open review interface does not seem to support changing the order.
>
> * We have specified September 2025 as the publication date.